# Estimation of Chlorophyll Content in Soybean Crop at Different Growth Stages Based on Optimal Spectral Index

**Hongzhao Shi** [1,2], **Jinjin Guo** [3,*], **Jiaqi An** [1,2], **Zijun Tang** [1,2], **Xin Wang** [1,2], **Wangyang Li** [1,2], **Xiao Zhao** [4], **Lin Jin** [1,2], **Youzhen Xiang** [1,2,*], **Zhijun Li** [1,2] **and Fucang Zhang** [1,2]

1. Key Laboratory of Agricultural Soil and Water Engineering in Arid and Semiarid Areas of Ministry of Education, Northwest A&F University, Yangling 712100, China
2. Institute of Water-Saving Agriculture in Arid Areas of China, Northwest A&F University, Yangling 712100, China
3. Faculty of Modern Agricultural Engineering, Kunming University of Science and Technology, Kunming 650500, China
4. School of Quality Education, Northwest A&F University, Yangling 712100, China
* Correspondence: 18292077095@163.com (J.G.); youzhenxiang@nwsuaf.edu.cn (Y.X.)

**Abstract:** Chlorophyll is an important component of crop photosynthesis as it is necessary for the material exchange between crops and the atmosphere. The amount of chlorophyll present reflects the growth and health status of crops. Spectral technology is a feasible method for obtaining crop chlorophyll content. The first-order differential spectral index contains sufficient spectral information related to the chlorophyll content and has a high chlorophyll prediction ability. Therefore, in this study, the hyperspectral index data and chlorophyll content of soybean canopy leaves at different growth stages were obtained. The first-order differential transformation of soybean canopy hyperspectral reflectance data was performed, and five indices, highly correlated with soybean chlorophyll content at each growth stage, were selected as the optimal spectral index input. Four groups of model input variables were divided according to the following four growth stages: four-node (V4), full-bloom (R2), full-fruit (R4), and seed-filling stage (R6). Three machine learning methods, support vector machine (SVM), random forest (RF), and back propagation neural network (BPNN) were used to establish an inversion model of chlorophyll content at different soybean growth stages. The model was then verified. The results showed that the correlation coefficient between the optimal spectral index and chlorophyll content of soybean was above 0.5, the R2 period correlation coefficient was above 0.7, and the R4 period correlation coefficient was above 0.8. The optimal estimation model of soybean and chlorophyll content is established through the combination of the first-order differential spectral index and RF during the R4 period. The optimal estimation model validation set determination coefficient ($R^2$) was 0.854, the root mean square error (RMSE) was 2.627, and the mean relative error (MRE) was 4.669, demonstrating high model accuracy. The results of this study can provide a theoretical basis for monitoring the growth and health of soybean crops at different growth stages.

**Keywords:** hyperspectral; soybean; chlorophyll content; optimal spectral index

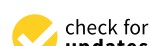



## 1. Introduction

Soybean is one of the most commonly used raw foods in daily life, and the national yield is closely related to both food security and people's quality of life [1]. Since 2012, China's soybean market has experienced dramatic fluctuations, soybean imports have increased significantly, and external dependence now exceeds 80% [2]. This poses a particular threat to national soybean food security, and, therefore, resolving the issues of low soybean yield and inconsistent quality is urgent.

The chlorophyll content is an important index to measure crop growth, which stems from the photosynthetic capacity of crop leaves [3]. It is the basis of material and energy exchange between crops and the external environment, and its value can reflect the health

and growth of crops. Traditional chlorophyll content measurement methods are destructive, leading to unreliable results [4]. Hyperspectral technology has the advantages of a high spectral resolution, being non-destructive and efficient, and possessing extensive information acquisition [5]. It can identify and analyze the spectral characteristics of a target object and is now increasingly used in the quantitative estimation of vegetation parameters and growth monitoring. Several spectral feature extraction methods have previously been proposed to effectively extract plant information in vegetation monitoring. The spectral index is the most commonly used variable parameter for establishing spectral estimation models, and combining it with various extraction methods for modeling has become a common research method [6]. Using hyperspectral technology to calculate and model the vegetation index can better monitor plant physiological parameters.

The combination of chlorophyll content measurement and hyperspectral technology provides a new idea for monitoring the dynamic change of chlorophyll content in crop leaves. Recent studies on the inversion of crop chlorophyll content by hyperspectral data have been undertaken. By observing the chlorophyll content and spectral characteristics of winter wheat at different growth stages, Liu et al. (2021) found that the chlorophyll content was significantly correlated with part of the spectral range and the remote sensing vegetation index; their model was highly accurate [7]. Yuan et al. (2017) collected chlorophyll content and hyperspectral data from leaves in the soybean flower bud differentiation stage and reversed them with mathematical models to obtain superior monitoring effects [8]. Most existing studies use the original crown height spectral reflectance data to construct mathematical inversion models of the chlorophyll content; however, the predictive effect is not ideal and the accuracy is low [9]. The integer order differential transformation method is introduced to process the original hyperspectral reflectance data to eliminate the spectral data background noise to a certain extent and improve the modeling accuracy [10]. Previous studies have inverted crop chlorophyll content by using the original hyperspectral data and the first-order differential transformation spectral data and found that compared with the initial data inversion results, the mathematical model established by using the first-order differential transformation data is more accurate, and the inversion results have a higher chlorophyll content prediction ability [11]. The traditional method uses the fixed band to construct the mathematical model, which cannot make good use of the spectral information contained in the spectral index [12]. The correlation matrix method selects the bands with high soybean chlorophyll content correlation to construct the optimal spectral index, which effectively solves the problem of the spectral characteristics being easily affected by the difference in physiological information of the crop itself [13] and significantly promotes the use of spectral data. Hyperspectral remote sensing technology divides the spectrum more precisely, allowing the best spectral index closely related to chlorophyll content to be selected in all available spectral bands.

Most previous studies have used differential transformation to process hyperspectral data to invert the physiological indices of a growth period of crops and fail to study the changes during the whole crop growth period in depth. Studying the whole crop growth period can better monitor the dynamic changes within the entire period and improve the predictive ability of physiological indicators. In this study, the soybean chlorophyll content during four different growth stages, i.e., the four-node stage (V4), complete flowering stage (R2), full pod stage (R4), and drum-grain stage (R6) was taken as the research object. A first-order differential transformation of the original hyperspectral reflectance was performed. The correlation matrix method was used to screen the bands with the highest soybean chlorophyll content correlation in the range of 350–1830 nm. Five spectral indices were selected, and a total of 20 optimal spectral indices were selected. On this basis, support vector machine (SVM), random forest (RF), and back propagation neural network (BPNN) were employed to construct the soybean chlorophyll content prediction model at different growth stages. The effects of different growth stages and machine learning methods on the accuracy of the soybean chlorophyll content prediction model were discussed, which

could provide a theoretical basis for monitoring the growth and health of soybean crops at different growth stages.

## 2. Materials and Methods

### 2.1. Research Area and Test Design

The soybean field experiment was conducted in the experimental field (loam) of the water-saving irrigation experimental station at the Key Laboratory of the Institute of Water-saving Agriculture in Arid Areas of China of Northwest A & F University. The test station coordinates are 108°24′ E, 34°20′ N, at an altitude of 524.7 m, which belongs to the warm temperate monsoon semi-humid climate zone where rainfall is concentrated from July to September, the average annual precipitation is 580 mm, and the average yearly temperature is 12.9 °C (Figure 1).

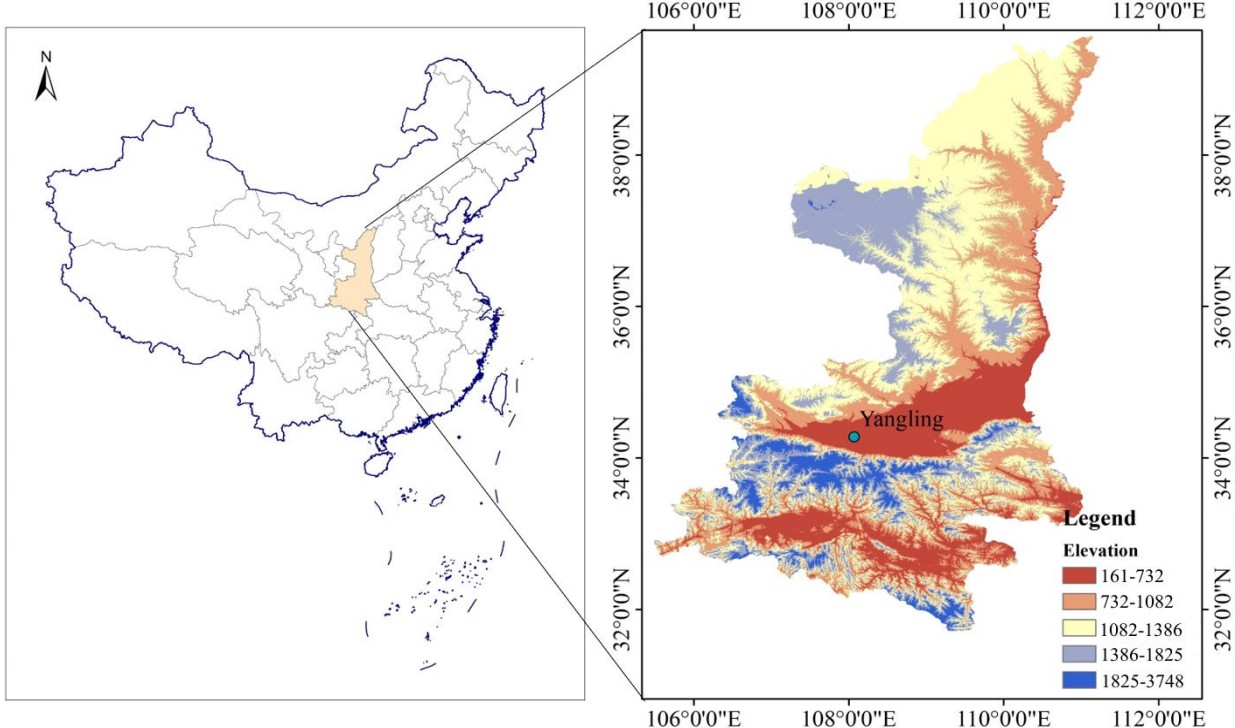

**Figure 1.** Study area.

Four nitrogen application levels were set in this experiment: N0: 0 kg/hm$^2$, N1: 60 kg/hm$^2$, N2: 120 kg/hm$^2$, N3: 180 kg/hm$^2$; four mulching treatments: SM: straw mulching, FM: agricultural film mulching, SFM: straw and agricultural film mulching, and NM: no mulching treatment. The amount of straw mulching was 6000 kg/hm$^2$. FM and SFM treatments utilized the ridging and covering film side sowing method. Two ridges with a width of 40 cm and a height of 25 cm were raised in the experimental plot, and the ridge surfaces were covered with 60 cm wide plastic film. Soybean was sown at 5 cm on the film side with a row spacing of 50 cm. N0 nitrogen treatment (CK) was the control. Each treatment was randomly arranged and repeated twice. The experimental area was 2.5 m × 6 m = 15 m$^2$, containing a total of 33 plots. Nitrogen, potassium, and phosphorus fertilizers were applied as base fertilizers before sowing. The test soybean variety, Shanning 17, was sown on 17 June 2021 and 9 June 2022, and harvested on 30 September 2021 and 28 September 2022. No significant pests or diseases occurred within the soybean growth period.

### 2.2. Data Collection

The chlorophyll content was measured at the four-node (V4), complete flowering (R2), full pod (R4), and drum-grain stages (R6) of the soybean. Chlorophyll content was measured using a SPAD-502 handheld chlorophyll meter. Since the SPAD-502 reading was closely related to the chlorophyll content, its value directly represented the chlorophyll content [14]. Ten soybean plants were randomly selected from each plot at different stages, and the SPAD values of the top 1, top 2, top 3, and top 4 leaves were measured from top to bottom along the main stem. At the same time, for the leaves at the four above leaf positions, from the base of the leaves, according to their length, every 1/3 was divided into intervals, which were defined as the base (B), middle (C), and top (R). The SPAD values were measured and the average values in the same experimental plots were calculated [14].

During the test period, spectral data were collected at different soybean growth stages. The test period was sunny, and the light was consistent. The spectral reflectance of the soybean canopy was measured by ASD Field-Spec 3 Analytical Spectral Devices, Inc., St, Boulder, CO, USA. The wavelength range of the instrument was 350–1830 nm. The spectral resolution of 350–1000 nm was 3 nm, and the sampling interval was 1.4 nm. The 1000–1830 nm resolution was 10 nm with a sampling interval of 2 nm, and the instrument automatically interpolates the sampling data to a 1 nm interval output, fiber length of 1.5 m, and field angle of 25° [15]. Before the hyperspectral data acquisition, the spectrometer was preheated and optimized, and the reference plate test and comparison were completed within 1 min. After the hyperspectral reflectance data acquisition of the previous test area, the reference plate correction was performed before the hyperspectral reflectance data acquisition of the latter test area. The crop canopy with balanced growth was selected in each experimental plot, and the experimenter held a spectral sensor probe. The optical fiber probe was placed vertically downward, about 1 m from the top of the canopy. In each plot, three quadrats representing the growth of the crop were selected for measurement. Each quadrat collected ten spectral curves each time, and the average value was used as the spectral reflectance of the quadrat. A total of 63 sets of data were collected for each growth stage, with a total of 252 sets of data collected during the experiment (Table 1).

**Table 1.** Statistics of chlorophyll content data at different growth stages of soybean.

| Indexes (Unitless) | Chlorophyll Content at Different Growth Stages (Unitless) | | | |
|---|---|---|---|---|
| | V4 | R2 | R4 | R6 |
| Sample size | 63 | 63 | 63 | 63 |
| Maximum values | 36.87 | 46.26 | 55.96 | 51.01 |
| Minimum values | 23.74 | 32.95 | 40.93 | 37.06 |
| Mean | 31.56 | 39.48 | 48.93 | 43.65 |
| Standard deviation | 3.08 | 3.71 | 4.00 | 3.51 |
| Coefficient of variation/% | 0.10 | 0.09 | 0.08 | 0.08 |

### 2.3. Spectral Data Processing and Spectral Index Construction

At the beginning, the binary Ostu algorithm and Canny edge detection algorithm were used to process the thermal infrared image in advance to eliminate the soil background. As the spectral data are disturbed by the environment (there is a strong water absorption band at 1350–1450 nm and an intense edge noise at 1801–1830 nm), this study was based on the reflectance of 351–1350 nm and 1451–1800 nm bands (a total of 1350 bands). To reduce (eliminate) the influence of background noise on the spectral reflectance curve, Savitzky-Golay convolution smoothing was used to preprocess the spectral data. The quadratic polynomial and number of smoothing points were 9, and function fitting and filtering denoising was performed [16]. At the four growth stages of the crop, the degree of crop senescence deepens; however, the field coverage was still good, so the spectrum was not affected by crop senescence and field residues.

Seven typical spectral indices were selected to fully utilize the information in the hyperspectral reflectance data, as shown in Table 2. Ratio Index (RI) and Triangular Vegetation Index (TVI) were strongly correlated with chlorophyll content and LAI of plants; however, the sensitivity decreased when the vegetation was dense [17]. The modified Simple Ratio (mSR) and modified Normalized Difference Index (mNDI) can optimize the specular emission effect of leaves and are sensitive to changes in leaves. Difference Index (DI), Normalized Difference Vegetation Index (NDVI), and Soil-Adjusted Vegetation Index (SAVI) can reflect the background influence of plant canopy and eliminate some radiation errors [18].

**Table 2.** The spectral index selected in this study.

| Spectral Index | Formula | Reference |
|---|---|---|
| Ratio Index (RI) | $R_i / R_j$ | [11] |
| Difference Index (DI) | $R_i - R_j$ | [11] |
| Modified Simple Ratio (mSR) | $R_i - R_{455} / R_j - R_{455}$ | [11] |
| Modified Normalized Difference Index (mNDI) | $R_i - R_j / R_i + R_j - 2R_{455}$ | [11] |
| Triangular Vegetation Index (TVI) | $0.5 \times \left( 120 \times (R_i - R_{550}) - 200 \times \left( R_j - R_{550} \right) \right)$ | [11] |
| Soil–Adjusted Vegetation Index (SAVI) | $(1 + 0.16)\frac{R_i - R_j}{R_i + R_j + 0.16}$ | [11] |
| Normalized Difference Vegetation Index (NDVI) | $R_i - R_j / R_i + R_j$ | [11] |

$R_i$ ($i$ = 1,2,3) is the reflectivity at any band, $R_{455}$ and $R_{550}$ are the hyperspectral reflectivity at 455 nm and 550 nm wavelengths.

Firstly, the spectral indices of all hyperspectral reflectance bands at each growth stage after first-order pretreatment were calculated by band-by-band spectral indices. Then, correlation analysis of the soybean chlorophyll content was performed using the correlation matrix method. The spectral indices of first-order were constructed with the *i* and *j* wavelengths, where the maximum correlation coefficient was located. In previous studies, the spectral reflectance at 445 nm and 550 nm wavelength positions was selected for the calculation of the spectral index, and good results were obtained [11]. Therefore, the spectral reflectance at 445 nm and 550 nm wavelength positions was also selected in this study.

*2.4. Model Construction and Verification*

Taking the four optimal spectral index combinations of different growth stages as input variables, three machine learning methods of SVM, RF, and BPNN were used to model and predict the chlorophyll content of soybean on MATLAB R2022a software. The ratio of the training set to the validation set was 2:1. The average value of the relevant results predicted by the machine learning model is the final model–fitting result in this experiment.

2.4.1. Model Construction

Support vector machine (SVM) is a binary classification machine learning algorithm with a Gaussian kernel and polynomial kernel as the basis kernel function, and a gradient descent algorithm is used to optimize the weight coefficient [19]. It has good generalization ability and robustness and has no over-fitting defect. It has been widely used in pattern recognition, classification, and small-sample regression analysis [20]. SVM uses Gaussian kernel and polynomial kernel as basis kernel function and adopts a gradient descent algorithm to optimize the weight coefficient [21].

Random forest (RF) is an integral model based on the idea of "Bagging"model. Because of its simple and convenient characteristics, it has been widely used in various regression and prediction problems. Since the RF model weighted the averages of the results of each tree to achieve the final output, the implementation of the model requires the construction

of a large number of decision trees and the construction of a set of decision trees by exchanging and changing covariates to improve the prediction performance [22]. Random forest needs to train N decision trees on N sampling training-test sets obtained by putting back samples. Based on the regression of the random forest model, the average prediction results of multiple decision trees are taken as the final results. When training-testing in the decision tree model, it is necessary to traverse each feature and each method to effectively determine the optimal number of decision trees. After error analysis and multiple training, the number of decision trees in the RF model was determined to be 500 [23].

Back propagation neural network (BPNN) is a multi-layer feedforward network trained by back propagation algorithm. It is essentially an iterative learning algorithm based on gradient descent. It has some defects, such as easily falling into a local minimum point, slow convergence speed, and possible oscillation when approaching the optimal solution [24]. The BPNN used in this study was provided by the Neural Network Toolbox in MATLAB. The transfer function of the hidden layer was set to TANSIG, and the Levenbeger-Marquardt (Train-LM) algorithm based on numerical optimization theory was used as the network training function. The number of neurons in the middle layer directly affects the simulation performance of the network. Thus, after several training sessions, we determined that the number of neurons in the middle layer was 10. In addition, during training, the maximum number of iterations was set to 1000 and the training target was set to $1 \times 10^{-5}$. After the neural network was trained, the test data were entered into the training network simulation to obtain the simulated values [25] (Figure 2).

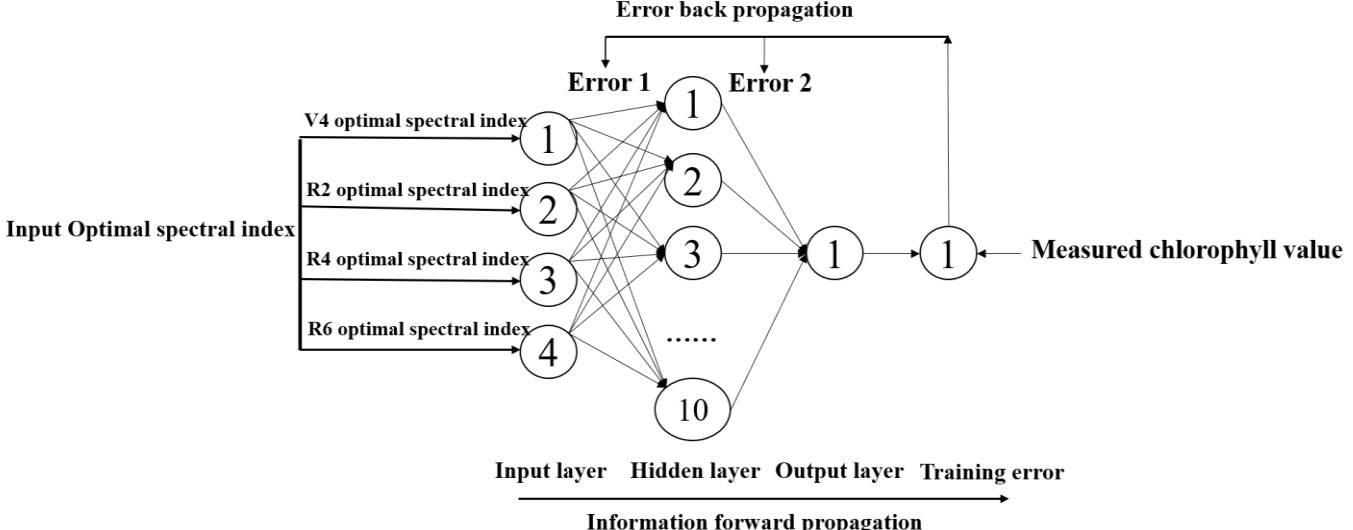

**Figure 2.** BPNN Structure Diagram in the experiments.

### 2.4.2. Model Verification

The model fitting results were evaluated by the determination coefficient ($R^2$), root mean square error (RMSE), and mean relative error (MRE) [26]. The larger the $R^2$, the more stable performance of the trained model, and the prediction results are more concentrated. The smaller RMSE and MRE indicate the higher prediction accuracy of the model [27]. The calculation formula is as follows:

$$R^2 = \frac{\sum_{i=1}^{n}(\hat{y}_i - \overline{y})^2}{\sum_{i=1}^{n}(y_i - \overline{y})^2} \tag{1}$$

$$RMSE = \sqrt{\frac{\sum_{i=1}^{n}(y_i - \overline{y})^2}{n}} \tag{2}$$

$$MRE = \frac{1}{n} \sum_{i=1}^{n} \frac{|\hat{y}_i - y_i|}{y_i} \times 100\% \tag{3}$$

In the formula $\hat{y}_i$—model prediction value;
$y_i$—actual sampling value;
$\bar{y}$—mean value;
$n$—sample number.

### 2.5. Data Analysis and Verification of the Prediction Accuracy of the Models

The experimental data were processed in Excel 2021 (Microsoft, Albuquerque, NM, USA). All graphics in the present paper were created using Origin Pro 2022 (Origin Lab Corp., Northampton, MA, US). For the diagrams of architectures used in the experiments, please refer to Figure 3.

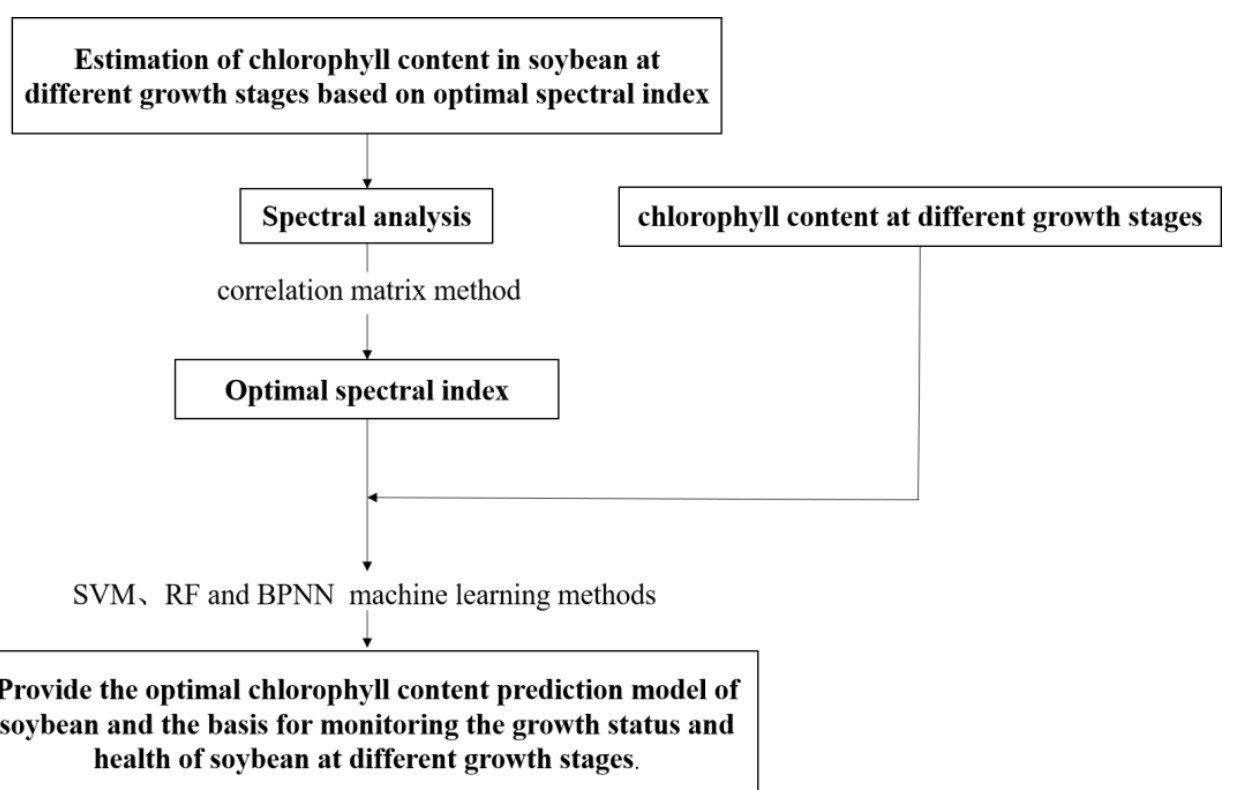

**Figure 3.** Diagrams of architectures used in the experiments.

### 3. Results

#### 3.1. Optimal Spectral Index Band Combination Extraction

Five indices with high soybean chlorophyll content correlation were selected from the seven spectral indices, as the optimal spectral index combination, and a correlation matrix diagram was drawn. As shown in Figures 4–6, from blue to red, the correlation between the spectral indices and soybean SPAD at each growth stage changes from a high negative correlation to a high positive correlation.

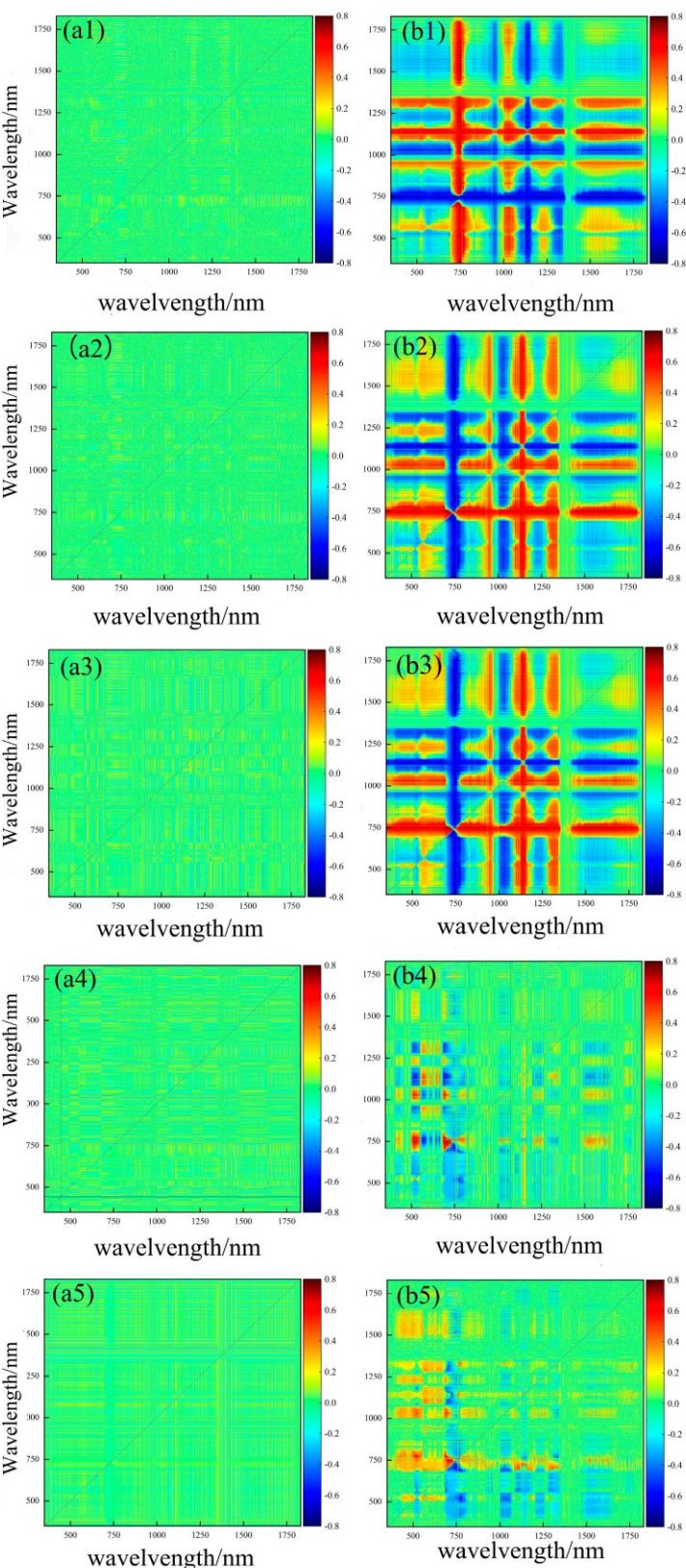

**Figure 4.** Correlation matrix diagram of spectral index and soybean chlorophyll content at different growth stages ((**a1**–**a5**): V4 growth period mNDI, NDVI, RI, mSR and SAVI and soybean chlorophyll content correlation matrix; (**b1**–**b5**): R2 growth period TVI, DI, SAVI, RI, and NDVI and soybean chlorophyll content correlation matrix).

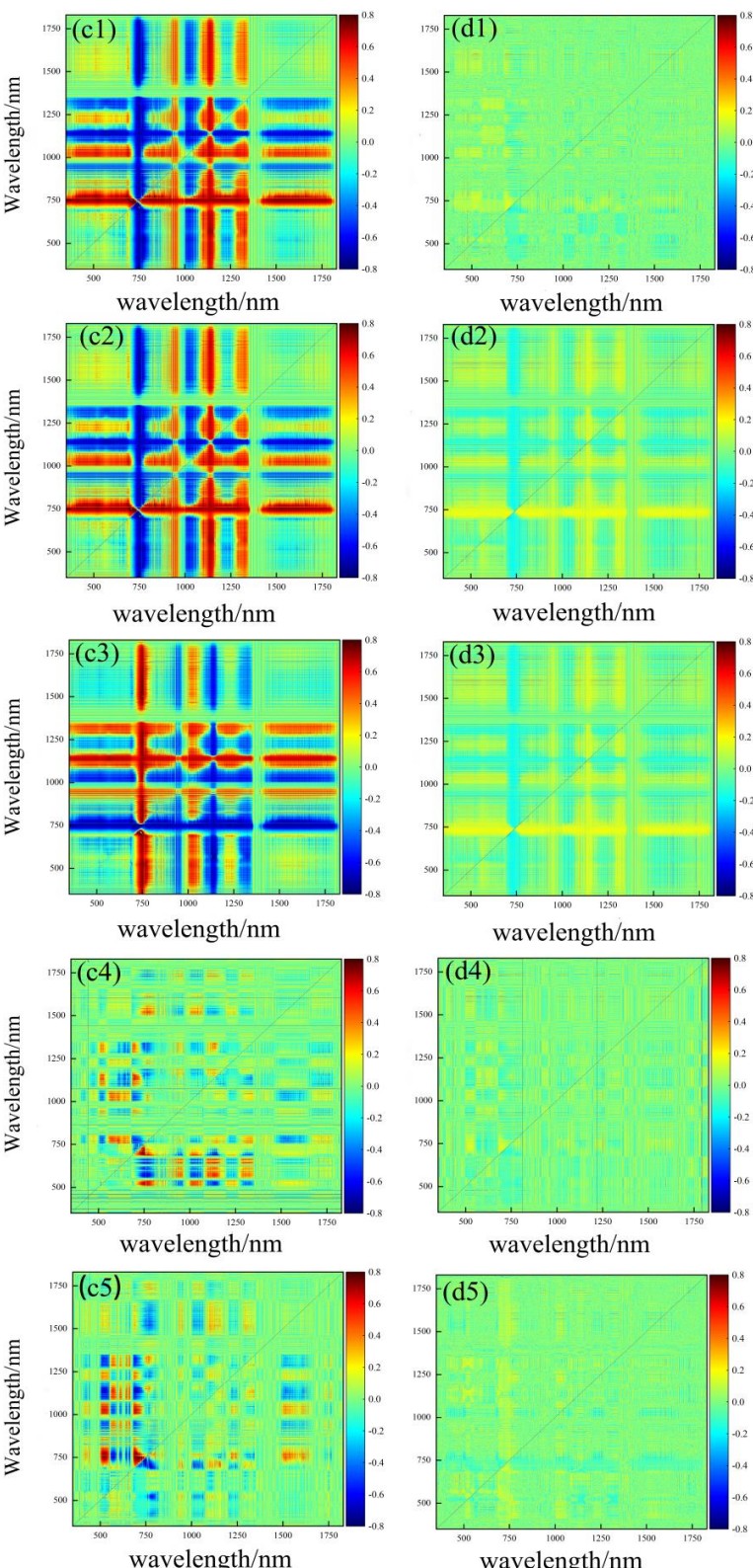

**Figure 5.** Correlation matrix diagram of spectral index and soybean chlorophyll content at different growth stages ((**c1–c5**): R4 growth period DI, SAVI, TVI, mSR, and RI and soybean chlorophyll content correlation matrix; (**d1–d5**): R6 growth period NDVI, SAVI, DI, RI and mNDI and soybean chlorophyll content correlation matrix).

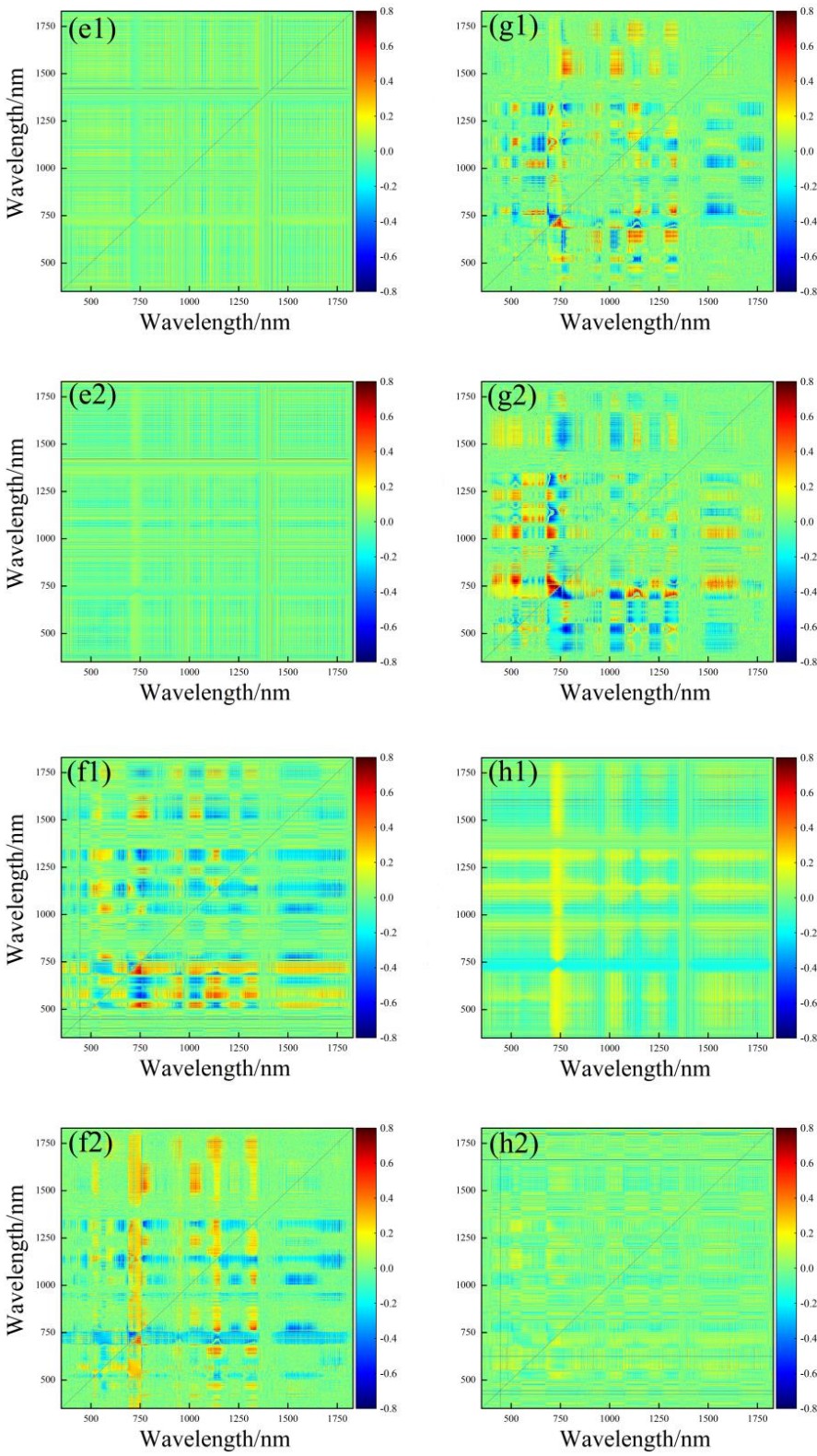

**Figure 6.** Correlation matrix diagram of spectral index and soybean chlorophyll content at different growth stages ((**e1,e2**): V4 growth period DI and TVI and chlorophyll content correlation matrix diagram; the correlation matrix of mSR and mNDI with chlorophyll content in (**f1,f2**): R2 growth period; (**g1,g2**): R4 growth period mNDI and NDVI and chlorophyll content correlation matrix; (**h1,h2**): R6 growth period TVI and mSR and chlorophyll content correlation matrix).

Figures 4–6 are the correlation matrix of the hyperspectral vegetation index and soybean SPAD in each growth period following first–order differential treatment. The correlation between the optimal spectral index and soybean chlorophyll content calculated by the first–order differential pretreatment of hyperspectral reflectance data was significantly improved. DI, TVI, and SAVI were highly correlated with soybean chlorophyll content in the first–order differential treatment. Although NDVI is often used to describe "greenness", the correlation of DI here is higher than NDVI. This is because DI may be more correlated with chlorophyll content at some stages due to factors such as the test environment and crop species. The correlation coefficients of V4 and R6 growth periods were low, at around 0.5–0.6. The correlation coefficient of the R2 growth period was higher, at about 0.7–0.8, and the correlation coefficient of the R4 growth period was the highest at above 0.8. The spectral index with the highest correlation coefficient with soybean chlorophyll content was DI in the R4 growth period, and the wavelength combination coordinate was (707,730). The ranked $r_{max}$ value of each spectral index and soybean SPAD value from high to low was DI > SAVI > TVI > mSR > RI > mNDI > NDVI. Among the above seven spectral indices, five, DI, SAVI, TVI, mSR, and RI, with high correlation coefficients, were selected as the optimal spectral index combinations. Their corresponding bands, (707,730), (713,752), (731,713), (739,710), and (713,752) were the optimal spectral index band combinations. The corresponding bands of the first–order differential optimal spectral index combination and the optimal spectral index combination in the remaining growth stages are presented in Table 3. The red edge is the band that reflects the physiological characteristics of plant growth and has the fastest green plant reflectance growth rate [28]. The 20 optimal spectral index bands selected in this experiment were all in the range of 670–760 nm and were all edge band range distributed in, or near, the red.

**Table 3.** Optimal vegetation index wavelength combinations at different growth stages.

| Growth Stages | Spectral Index | $r_{max}$ | Wavelength Position (i,j)/(nm) | Optimal Vegetation Index |
|---|---|---|---|---|
| V4 | mNDI | 0.541 | (759,680) | mNDI, NDVI, RI, mSR, SAVI |
| | NDVI | 0.531 | (725,714) | |
| | RI | 0.521 | (720,706) | |
| | mSR | 0.506 | (756,689) | |
| | SAVI | 0.500 | (660,676) | |
| | DI | 0.500 | (660,676) | |
| | TVI | 0.472 | (676,689) | |
| R2 | TVI | 0.777 | (755,691) | TVI, DI, SAVI, RI, NDVI |
| | DI | 0.757 | (710,734) | |
| | SAVI | 0.756 | (721,764) | |
| | RI | 0.748 | (691,755) | |
| | NDVI | 0.734 | (729,747) | |
| | mSR | 0.715 | (752,691) | |
| | mNDI | 0.695 | (741,707) | |
| R4 | DI | 0.832 | (707,730) | DI, SAVI, TVI, mSR, RI |
| | SAVI | 0.831 | (713,752) | |
| | TVI | 0.825 | (731,713) | |
| | mSR | 0.801 | (739,710) | |
| | RI | 0.800 | (713,752) | |
| | mNDI | 0.797 | (726,713) | |
| | NDVI | 0.794 | (728,752) | |
| R6 | NDVI | 0.538 | (705,718) | NDVI, SAVI, DI, RI, mNDI |
| | SAVI | 0.520 | (707,723) | |
| | DI | 0.520 | (701,711) | |
| | RI | 0.519 | (705,688) | |
| | mNDI | 0.513 | (723,691) | |
| | TVI | 0.503 | (708,692) | |
| | mSR | 0.496 | (675,705) | |

### 3.2. Construction and Comparison of Soybean Chlorophyll Prediction Models at Different Growth Stages

The optimal spectral index combination of different growth stages was the independent variable, and the soybean leaf chlorophyll content was the response variable. SVM, RF, and BPNN methods were used to construct soybean chlorophyll estimation models at different growth stages. The model accuracy was comprehensively evaluated from three aspects of $R^2$, RMSE, and MRE. The predictions of different modeling methods for the soybean leaf areas are shown in Table 4.

**Table 4.** Comparison of model prediction accuracy evaluation under different growth stages.

| Growth Stages | Evaluation Indicators | SVM | | RF | | BPNN | |
|---|---|---|---|---|---|---|---|
| | | Modeling Set | Validation Set | Modeling Set | Validation Set | Modeling Set | Validation Set |
| V4 | $R^2$ | 0.551 | 0.694 | 0.730 | 0.726 | 0.576 | 0.674 |
| | RMSE | 1.987 | 1.809 | 1.617 | 2.192 | 1.935 | 2.129 |
| | MRE | 4.837 | 4.891 | 4.070 | 4.373 | 5.033 | 5.290 |
| R2 | $R^2$ | 0.651 | 0.645 | 0.776 | 0.746 | 0.731 | 0.726 |
| | RMSE | 2.124 | 2.496 | 1.684 | 2.149 | 1.863 | 2.221 |
| | MRE | 4.104 | 4.930 | 3.507 | 4.746 | 3.676 | 4.353 |
| R4 | $R^2$ | 0.734 | 0.717 | 0.862 | 0.854 | 0.754 | 0.743 |
| | RMSE | 1.995 | 2.578 | 1.458 | 2.627 | 1.945 | 2.508 |
| | MRE | 2.882 | 4.458 | 2.259 | 4.669 | 3.008 | 4.573 |
| R6 | $R^2$ | 0.516 | 0.587 | 0.724 | 0.718 | 0.512 | 0.684 |
| | RMSE | 2.310 | 2.706 | 1.804 | 2.884 | 2.353 | 2.190 |
| | MRE | 4.158 | 5.487 | 3.504 | 5.078 | 4.084 | 3.886 |

The corresponding soybean chlorophyll estimation model $R^2$ was R4 > R2 > R6 > V4 in different soybean growth stages. The RMSE and MRE values were small, which can be expressed in the order of R6 > R2 > V4 > R4. In the R4 growth period, the $R^2$ of the modeling and validation set of the chlorophyll content estimation model, constructed by the RF method, exceeded 0.8. The $R^2$ of the modeling set and validation set of the chlorophyll content estimation model, built by the SVM and BPNN methods, exceeded 0.7, both of which had an excellent nonlinear fitting ability. In the same growth period, the accuracy of the modeling and verification sets of the soybean chlorophyll content estimation model, constructed by the three modeling methods, were as follows: RF > BPNN > SVM, indicating that RF was the optimal model construction method. The practical information on soybean chlorophyll content in the optimal spectral index combination can therefore be further extracted. In summary, the R4 growth period is the best growth period for chlorophyll content prediction, and the RF method is the best model construction method. The $R^2$ of the optimal soybean chlorophyll content estimation model modeling and verification sets were 0.862 and 0.854, RMSE was 1.458 and 2.627, and MRE was 2.259 and 4.669 (Figure 7).



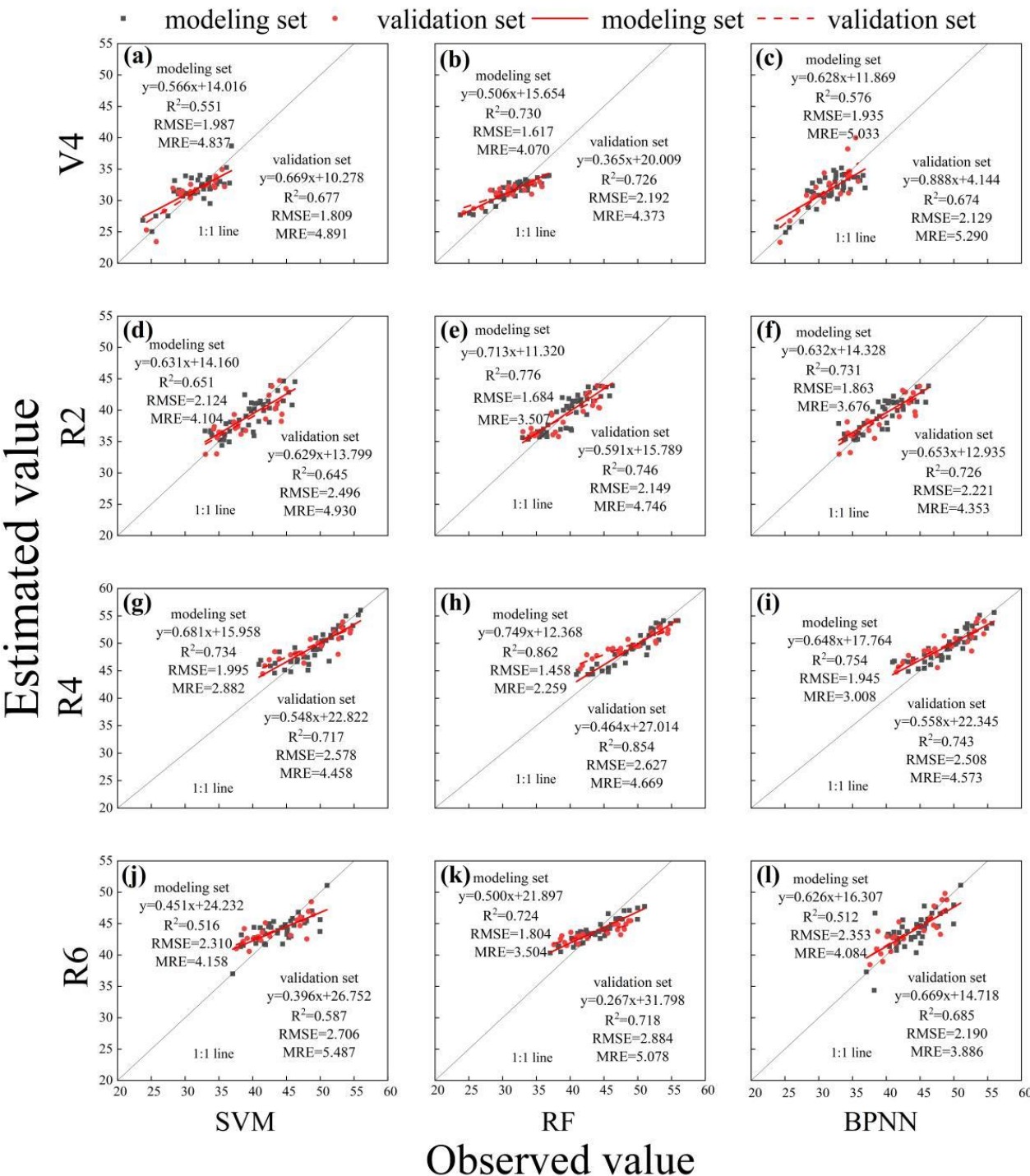

**Figure 7.** Model evaluation results, (**a**–**c**) is the prediction model of soybean chlorophyll content in the V4 growth period constructed by SVM, RF, and BPNN methods; (**d**–**f**) is the result of the R2 growth period soybean chlorophyll content prediction model created by SVM, RF, and BPNN; (**g**–**i**) is the result of R4 growth period soybean chlorophyll content prediction model constructed by SVM, RF, and BPNN; (**j**–**l**) is the result of R6 growth period soybean chlorophyll content prediction model produced by SVM, RF, and BPNN methods.

## 4. Discussion

It is found that the model's prediction ability, built by directly using the original hyperspectral reflectance for the crop chlorophyll content inversion model, is often moderate [29]. The introduction of differential transformation processing reduces the

spectral reflectance background noise, highlights the hyperspectral feature information, enhances the correlation between the hyperspectral reflectance and crop chlorophyll content [30,31], and promotes the inversion model accuracy. This study found that the inversion model of the soybean chlorophyll content at different growth stages was constructed by using soybean hyperspectral reflectance data under the first–order differential order treatment. Consequently, the correlation between the spectral index and soybean chlorophyll content was high, as was the model accuracy. This is because the first–order differential spectral index contains more useful spectral information related to chlorophyll content, and chlorophyll content monitoring is more accurate. Therefore, it is feasible to invert crop chlorophyll content by using the first–order differential spectral index. The optimal spectral index, constructed by screening bands in the whole band range, contained more practical information on soybean chlorophyll content. The strong absorption of chlorophyll in the red band and the strong reflection of the near–infrared band inside the leaves make the red–edge band the fastest growth of green plant reflectance and the most important indicator band that can reflect the physiological characteristics of plant growth. The red edge band is a vital indicator to describe plant growth, and more than 80% of physical and chemical plant parameters can be mapped from the spectral information contained in the red edge band [32,33]. The bands corresponding to the 20 optimal spectral indices created in this study are distributed in the red edge or near–infrared band. This may be because the red edge is highly sensitive to chlorophyll content and describes the absorption capacity of the plant chlorophyll to red light and near–infrared radiation [34,35]. The change of SPAD is closely related to the crop chlorophyll content and photosynthesis [36]; therefore, the red edge band strongly correlates with the soybean chlorophyll content.

In this study, the soybean leaves were lusher during the R4 growth period and the leaves contained more chlorophyll, and therefore had more useful chlorophyll spectral information, resulting in a superior chlorophyll content prediction ability [37]. In the R4 growth stage, the correlation between the spectral index and chlorophyll content was the highest, and the chlorophyll content was closely related to the yield, showing a positive correlation. This can provide a theoretical basis for farmers to predict yield in crop production, i.e., the R4 period can be the best period for farmers to predict yield. In Figure 7, the observed chlorophyll values are often higher than the estimated values. Because the experiment is inevitably affected by errors, the deviation of the instrument and the measurement angle may affect the results. In addition, differences in the field of view between the SPAD meter (leaf level) and the ASD (plant level) can also lead to differences between observations and estimates. Among the three methods, the RF method–based prediction model was the most accurate, indicating that RF has a greater ability to extract chlorophyll content–related information from spectral reflectance data. This is because RF algorithm training can be highly parallelized and is not sensitive to some missing features. The trained model has a minor variance, strong generalization ability, higher background noise and outlier tolerance, which makes it more suitable for solving nonlinear problems [38]. BPNN is one of the most widely used neural networks. Through iterative processing, the network weights of connected neurons are continuously adjusted to minimize the error between the final output and the expected result [39]. However, the neural network model's accuracy is easily reduced due to the algorithm falling into the local extremum, the weight converging to the local minimum point, and its slow convergence speed [40]. In this study, the BPNN model accuracy was lower than that of the RF model, which may be due to the relatively small number of samples but the high amount of model training, resulting in a decrease in model accuracy and generalization ability [41]. Compared with RF and BPNN, the SVM model is poor at chlorophyll content prediction. SVM has weak anti–interference ability and there is no standard for the kernel function selection of inseparable linear data. It is sensitive to missing and noise data and does not support feature diversity [42]. The results of this study show that the best fitting accuracy is obtained when using the RF method to construct the chlorophyll content prediction model with the first–order differential optimal spectral index combination of each growth period as the input variable. This can provide a

reference for accurately predicting soybean chlorophyll content at different growth stages and further exploring the dynamic changes of hyperspectral reflectance at each growth stage. In addition, there are some applications of new machine learning methods. Shah et al. found that tabu data learning architecture (TabNet) can be used for hyperspectral image classification. Experimental results obtained on different hyperspectral datasets demonstrate the superiority of the proposed approaches in comparison with other state–of–the–art techniques including DNNs and decision tree variants [43]. He et al. combined hyperspectral imaging with machine learning to characterize banana samples with hyperspectral images. It was found that the XGBoost model can 100% correctly distinguish natural and artificial ripening bananas compared with models such as support vector machine (SVM) and multi–layer perceptron (MLP). This method can be used to classify different ripening bananas quickly and non–invasively [44]. This provides a reference for the future use of new machine learning methods to estimate physiological indicators of crops.

At present, the use of hyperspectral data to study the model of LAI, biomass, nitrogen, and chlorophyll of crops has achieved good results; however, the commonly used ground hyperspectral data can only be obtained on the ground, and there are many restrictions. In the future, it will be necessary to test and improve the model through experiments in different regions, different scales, and different varieties, as well as other machine learning methods, to realize the effective unification of model estimation accuracy and universality, and provide a reference for predicting the chlorophyll content of soybean crops in multiple growth periods by using multi–source remote sensing methods such as multi–spectral and hyperspectral.

## 5. Conclusions

This study optimized hyperspectral reflectance following first–order differential transformation by the correlation matrix method over the whole band range. It was found that the constructed optimal spectral index was highly correlated with the soybean chlorophyll content at different growth stages, and the correlation with the soybean chlorophyll content during the R4 growth stage was the highest. The accuracy of the chlorophyll content prediction model constructed via the RF method was significantly higher than that of the SVM and BPNN methods. The R4 growth period was the best growth period for chlorophyll content prediction, and the RF method was the best model construction method.

**Author Contributions:** H.S.: Methodology, Formal analysis, Writing—original draft, Visualization. X.W. and Z.T.: Methodology, Software. J.A. and J.G.: Data curation, Writing—review and editing, Formal analysis. Y.X.: Supervision, Writing—review and editing, Funding acquisition. X.Z., L.J., W.L. and Z.L.: Data curation. F.Z.: Supervision, Project administration. All authors have read and agreed to the published version of the manuscript.

**Funding:** This study was supported by the National Natural Science Foundation of China (No. 52179045).

**Data Availability Statement:** Not applicable.

**Conflicts of Interest:** The authors declare no conflict of interest.

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
