# Peer review of "Estimation of Chlorophyll Content in Soybean Crop at Different Growth Stages Based on Optimal Spectral Index"

_agronomy, doi:10.3390/agronomy13030663_

Round 1

Reviewer 1 Report

The authors compared the abilities of three machine learning algorithms (support vector machine, random forest, and back propagation neural network) and seven types of vegetation indices to predict soybean chlorophyll content across four soybean growth stages. They used a SPAD meter to measure chlorophyll levels, and an ASD FieldSpec 3 to collect hyperspectral data for chlorophyll prediction. The authors found random forest outperformed the other two methods using the first differential transformation of spectra, the difference index with reflectances at 707 nm and 730 nm, in the full-pod growth stage. Questions and comments follow:

1)     In L63, what does “part of the spectral index” mean? Is this supposed to say “part of the spectral range”?

2)     The SPAD-502 meter uses optical densities at 650 nm and 940 nm to estimate chlorophyll, so it is interesting indices using those wavelengths did not have higher correlation coefficients.

3)     In L136, what is “cell”? Is it the same as “interval”?

4)     In L138, what is “ordered day”?

5)     In L139, is “ASD, Inc” now “Malvern Panalytical”?

6)     Since SPAD meter chlorophyll measurements were collected at the leaf level, would it not make more sense to collect ASD spectra with a 1-degree attachment rather than the bare fiber optic with a 25-degree FOV?

7)     Please check the capitalization of section headings and figure captions for consistency.

8)     For SVM, why was the radial basis function not used, since it can handle non-linearity better?

9)     For BPNN, what would happen if more layers were incorporated into the model?

10)  A diagram of the BPNN architecture might be more informative than the current diagram in Figure 2.

11)  For the spectral indices listed in Table 2, please elaborate on why the reflectances at 455 nm and 550 nm were used.

12)  Also in Table 2, there are so many chlorophyll-specific indices (Gitelson Chl index, Modified Chlorophyll Absorption in Reflectance Index, Triangular Greenness Index, Normalized Pigment Chlorophyll Index, etc). Why were these not included?

13)  It is interesting that DI outperformed NDVI since the latter is so often used to describe “greenness.” Please elaborate on why this might have happened.

14)  In Table 3, some of the band pairs have bands that are probably autocorrelated, like NDVI using reflectances at 715 nm and 714 nm. How might this be affecting your results?

15)  For Figures 3-6, the subfigure labels are too small.

16)  Please reword the sentence in L257-259.

17)  Since correlations were strongest in R4, is this also the best stage for farmers to estimate soybean yield?

18)  In Table 4, I suggest changing “Period of duration” to “Growth Stage” for consistency. If possible, the table needs to be made wider (or in landscape) to avoid the multiple rows needed for the column headings.

19)  Also looking at Table 4, the correlations for modeling sets are sometimes substantially higher than for validation sets. This indicates to me the models might not be completely optimized.

20)  In Figure 6, it would be helpful to have row labels (growth stages) and column labels (algorithm).

21)  Also in Figure 6, we see observed chlorophyll values are often higher than estimated values. This might be because of the differences in field of view of the SPAD meter (leaf level) and the ASD (plant level).

22)   Please describe the canopy closure and/or senescent levels in each of the growth stages. How much influence might there be from the soil background and/or dead vegetation?

23)  Usually, information on crop vigor is obtained by comparing reflectances at the red and near-infrared bands, rather than only the red edge. Please elaborate on why the red edge bands were most important in this study.

24)  In L314, does “this growth period” refer to R4?

25)  In L321, do you mean “higher background noise” tolerance?

26)  There are many ways to avoid local extrema (appropriate learning rates) and the detrimental effects of small sample sizes (sample transformations) in neural networks. If these were implemented, would correlations be higher than those achieved using RF? Also consider testing the radial basis kernel function in SVM, as that might also provide better results than RF.

27)  Please make data publicly available as they provide crucial reference data for such model-building efforts.

Author Response

Responses to Reviewer #1 (Manuscript ID: Agronomy- 2199247)

Thank you for addressing all my comments.

Thank you for your careful review and positive comments. We have now incorporated the reviewer’s comments and suggestions in preparation of the revised manuscript. The modified part is marked in red in the manuscript.

(1)Evaluation experts proposed:“ In L63, what does ‘part of the spectral index’ mean? Is this supposed to say ‘part of the spectral range’?”

A : Thank you for pointing this out, we have revised as“part of the spectral range”.

Line64:Liu et al. (2021) found that the chlorophyll content was significantly correlated with part of the spectral range and the remote sensing vegetation index, their model was highly accurate.

(2)Evaluation experts proposed:“The SPAD-502 meter uses optical densities at 650 nm and 940 nm to estimate chlorophyll, so it is interesting indices using those wavelengths did not have higher correlation coefficients.”

A: Thank you for pointing this out. The optimal band used in some indexes in this paper is the band near 650 nm, which has a certain deviation. After the first-order differential transformation, the results also have corresponding changes, so the correlation coefficient is not so high.

(3)Evaluation experts proposed:“In L136, what is “cell”? Is it the same as ‘interval’?”

A:Thank you for pointing this out, we have revised as“experimental plots”.

Line139:The SPAD values were measured and the average values in the same experimental plots were calculated.

(4)Evaluation experts proposed:“  In L138, what is ‘ordered day’?

A:Thank you for pointing this out,we have revised as“During the test period”.

Line141: During the test period, spectral data was collected at different soybean growth stages.

(5)Evaluation experts proposed:“  In L139, is “ASD, Inc” now ‘Malvern Panalytical’?”

A:Thank you for pointing this out, the hyperspectral measuring instrument was produced by ASD ( Analytical Spectral Devices, Inc., Boulder, CO, USA ).

(6)Evaluation experts proposed:“ Since SPAD meter chlorophyll measurements were collected at the leaf level, would it not make more sense to collect ASD spectra with a 1-degree attachment rather than the bare fiber optic with a 25-degree FOV?”

A:Thank you for pointing this out, in the growth period of soybean, when the field angle is 25 °, the measurement height of 100 cm can better obtain the hyperspectral reflectance of soybean. In the current experimental environment and measurement height, the 25-degree optical fiber probe can collect the spectral data of crop leaves more easily and comprehensively, which is more suitable for my research content.

(7)Evaluation experts proposed:“ Please check the capitalization of section headings and figure captions for consistency.”

A:Thank you for pointing this out,we checked the case of section title and chart title and modified the case to be consistent.

(8)Evaluation experts proposed:“For SVM, why was the radial basis function not used, since it can handle non-linearity better?”

A:Thank you for pointing this out,SVM uses Gaussian kernel and polynomial kernel as basis kernel function, Gaussian kernel function can map a sample to a higher dimensional space, the kernel function is the most widely used, whether large sample or small sample has better performance ; the polynomial kernel makes the original linear inseparable data linearly separable, which can solve the nonlinear problem, and has good adaptability and application results for the kernel function in this study.

(9)Evaluation experts proposed:“For BPNN, what would happen if more layers were incorporated into the model?”

A:Thank you for pointing this out,For BPNN, the more layers, the smaller the error of the whole network, but it will complicate the whole network, increase the training time of the network, and may also appear ' over fitting ' phenomenon. In this study, after multiple model training, it is found that the current number of layers is the best, which can achieve better results.

(10)Evaluation experts proposed:“ A diagram of the BPNN architecture might be more informative than the current diagram in Figure 2.”

A:Thank you for pointing this out,the BPNN structure diagram has been added in the paper.

(11)Evaluation experts proposed:“For the spectral indices listed in Table 2, please elaborate on why the reflectances at 455 nm and 550 nm were used.”

A:Thank you for pointing this out,in previous studies, the spectral reflectance at 445 nm and 550 nm wavelength positions was selected for the calculation of spectral index, and good results were obtained. Therefore, the spectral reflectance at 445 nm and 550 nm wavelength positions was also selected in this study.

Román J R, Rodríguez-Caballero E, Rodríguez-Lozano B,et al. Remote Sensing, 2019,11(11):1.

Sun H, Feng M C, Xiao L J, et al.PLOS ONE,2019,14(6):e0216890.

Lu J Z, Ehsani R, Shi Y Y.Scientific Reports,2018,8:2793.

Liu Shuang, Yu Haiye, Zhang Junhe, Zhou Haigen, Kong Lijuan, Zhang Lei, Dang Jingmin, Sui Yuanyuan. Research on inversion model of soybean leaf chlorophyll content based on optimal spectral index [ J ].Spectroscopy and spectral analysis, 2021,41 ( 06 ) : 1912-1919.

(12)Evaluation experts proposed:“Also in Table 2, there are so many chlorophyll-specific indices (Gitelson Chl index, Modified Chlorophyll Absorption in Reflectance Index, Triangular Greenness Index, Normalized Pigment Chlorophyll Index, etc). Why were these not included?”

A:Thank you for pointing this out,Ratio Index ( RI ), Difference Index ( DI ), Modified Simple Ratio ( mSR ), Modified Normalized Difference Index ( mNDI ), Triangular Vegetation Index ( TVI ), Soil-Adjusted Vegetation Index ( SAVI ) and Normalized Difference Vegetation Index ( NDVI ) are seven typical spectral indices related to chlorophyll content. In addition, it has been widely used in previous studies, and the effect is better, so this study also uses these indexes, and will consider the substitution of other indexes in future studies.

(13)Evaluation experts proposed:“It is interesting that DI outperformed NDVI since the latter is so often used to describe “greenness.” Please elaborate on why this might have happened.”

A:Thank you for pointing this out,Although NDVI is often used to describe“greenness.”, DI may be more correlated with chlorophyll content at some stages due to factors such as the test environment and crop species.

(14)Evaluation experts proposed:“In Table 3, some of the band pairs have bands that are probably autocorrelated, like NDVI using reflectances at 715 nm and 714 nm. How might this be affecting your results?”

A:Thank you for pointing this out,after we checked, we found that the data of the wave segment position in the table had some errors in the input, and the error part had been corrected.

(15)Evaluation experts proposed:“For Figures 3-6, the subfigure labels are too small.”

A:Thank you for pointing this out,we have enlarged the label in the figure to the appropriate size.

(16)Evaluation experts proposed:“ Please reword the sentence in L257-259.”

A:Thank you for pointing this out,we have reword the sentence in L257-259.

(17)Evaluation experts proposed:“Since correlations were strongest in R4, is this also the best stage for farmers to estimate soybean yield?”

A:Thank you for pointing this out,in the R4 growth stage, the correlation between spectral index and chlorophyll content was the highest, and the chlorophyll content was closely related to the yield, showing a positive correlation. This study can provide a theoretical basis for farmers to predict yield in crop production, that is, R4 period can be the best period for farmers to predict yield.

(18)Evaluation experts proposed:“In Table 4, I suggest changing ‘Period of duration’ to ‘Growth Stage ’for consistency. If possible, the table needs to be made wider (or in landscape) to avoid the multiple rows needed for the column headings.”

A:Thank you for pointing this out,we have changed  “Period of duration” to “Growth Stage”,and make the table wider.

(19)Evaluation experts proposed:“Also looking at Table 4, the correlations for modeling sets are sometimes substantially higher than for validation sets. This indicates to me the models might not be completely optimized.”

A:Thank you for pointing this out,The model has been checked and optimized, and the corresponding parts of the text have been changed.

(20)Evaluation experts proposed:“In Figure 6, it would be helpful to have row labels (growth stages) and column labels (algorithm).”

A:Thank you for pointing this out,we have added row labels (growth stages) and column labels (algorithm).

(21)Evaluation experts proposed:“Also in Figure 6, we see observed chlorophyll values are often higher than estimated values. This might be because of the differences in field of view of the SPAD meter (leaf level) and the ASD (plant level).”

A:Thank you for pointing this out,In Fig.6, the observed chlorophyll values are often higher than the estimated values. The influence of multiple factors in the experiment is inevitably affected by errors. The deviation of instrument and measurement angle may affect the results. In addition, as experts said, the different field of vision of SPAD measuring instrument ( leaf level ) and ASD ( plant level ) will also lead to differences between the observed value and the estimated value.

(22)Evaluation experts proposed:“Please describe the canopy closure and/or senescent levels in each of the growth stages. How much influence might there be from the soil background and/or dead vegetation?”

A:Thank you for pointing this out,in the four-node stage(V4), Soybean is in the earlier growing stage, the vegetation area is moderate, and a small amount of soil is exposed. In complete flowering stage (R2), full pod stage (R4) and drum-grain stage (R6),The vegetation is dense and almost completely covers the ground.The degree of aging gradually deepened.The soil background was eliminated in the spectral treatment, which had little effect on the results. In addition, due to better field management, the crop grew well, no pests and diseases, almost no dead vegetation, so it did not affect the experimental results.

(23)Evaluation experts proposed:“Usually, information on crop vigor is obtained by comparing reflectances at the red and near-infrared bands, rather than only the red edge. Please elaborate on why the red edge bands were most important in this study.”

A:Thank you for pointing this out,the strong absorption of chlorophyll in the red band and the strong reflection of the near-infrared band inside the leaves make the red-edge band the fastest growth of green plant reflectance and the most important indicator band that can reflect the physiological characteristics of plant growth. More than 80 % of the physical and chemical parameters of plants can be mapped from the spectral information contained in the red-edge band. Therefore, the red edge band is the most important in this study.

(24)Evaluation experts proposed:“In L314, does ‘this growth period’ refer to R4?”

A:Thank you for pointing this out,“this growth period” is refer to R4,I have changed it to “R4 growth period”.

Line351:In this study, the soybean leaves are lusher during R4 growth period and the leaves contain more chlorophyll.

(25)Evaluation experts proposed:“ In L321, do you mean “higher background noise” tolerance?”

A:Thank you for pointing this out, In L321,we mean “higher background noise”,we have changed it to “higher background noise tolerance”.

Line358: The trained model has a minor variance, strong generalization ability, higher background noise and outlier tolerance,  which makes it more suitable for solving nonlinear problems

(26)Evaluation experts proposed:“There are many ways to avoid local extrema (appropriate learning rates) and the detrimental effects of small sample sizes (sample transformations) in neural networks. If these were implemented, would correlations be higher than those achieved using RF? Also consider testing the radial basis kernel function in SVM, as that might also provide better results than RF.”

A:Thank you for pointing this out,Using other methods in the neural network can avoid local extremum ( appropriate learning rate ) and the harmful effects of small sample size ( sample conversion ) or consider testing the radial basis kernel function in SVM, which may provide better results than RF. I will consider the above factors in the next study, use other methods to avoid harmful effects and provide better treatment methods.

(27)Evaluation experts proposed:“Please make data publicly available as they provide crucial reference data for such model-building efforts.”

A:Thank you for pointing this out,due to the limited nature of the study, I will disclose the corresponding data after the employment of this paper.

Reviewer 2 Report

This paper reports the accurate estimation of chlorophyll content in soybean at different growth stages using the spectral analysis method by selecting the optimal spectral index.

However, I have some doubts about this paper, which are as follows.

1. In the title, is “soybean crop” more appropriate than “soybean”?

2. In lines 89-91, “i.e., the four-node stage (V4), complete flowering stage (R2), full pod stage (R4), and drum-grain stage (R6) was taken as the research object.”

Why take these four growth stages as the research object? Are chlorophyll content estimates for other growth stages of soybean not meaningful?

3. In lines 113-114 and other places, please note the space between the number and unit.

4. In lines 128-129, I know that SPAD values correlate highly with crop chlorophyll content. However, the SPAD value cannot completely equal to the chlorophyll content. Why not use chemical methods (such as extraction) to determine directly the chlorophyll content of soybean leaves?

5. In lines 147-149, “A total of 63 sets of data were collected for each growth stage, with a total of 252 sets of data collected during the experiment…”

How to get the 63 sets of data for each growth stage? Please explain it in detail.

6. In lines 154-156, “… Savitzky-Golay convolution smoothing was used to preprocess the spectral data.”

Can S-G smoothing alone complete so many preprocessing functions, including background noise, baseline drift, and stray light?

Please consider other preprocessing methods, such as SNV.

7. In lines 178-180. “The larger the R2, the higher the prediction accuracy of the model. The smaller RMSE and MRE indicate that the model's performance is more stable, and the prediction results are more concentrated.”

I do not think your description is correct.

The large R2 value indicates the more stable performance of the trained model. The smaller RMSE and MRE indicate the higher prediction accuracy of the model.

8. In Figure 2, there is a problem with the structure of the architecture diagram. “chlorophyll content at different growth stages” should not be behind the “Optimal spectral index”, and should be in parallel with “Spectral analysis”.

9. In lines 197-207, Table 2 and the corresponding description content should be placed in Section 2 Materials and methods.

10. In line 206, “Figures 1 and 2” are not correct here.

11. In line 268, “… both of which had an excellent linear fitting effect.”

Do SVM and BPNN methods have excellent linear fitting capabilities? Isn't it the nonlinear fitting ability?

12. In Figure 6, please note the number of decimal places.

13. In the Results and Discussion section, the paper emphasizes that the first-order differential spectra contain more useful information, and the best spectral index is selected from the first-order differential spectra.

However, this paper does not perform a comparison between the original spectrum and the first-order differential spectrum.

Author Response

Responses to Reviewer #2 (Manuscript ID: Agronomy- 2199247)

Thank you for addressing all my comments.

Thank you for your careful review and positive comments. We have now incorporated the reviewer’s comments and suggestions in preparation of the revised manuscript. The modified part is marked in red in the manuscript.

(1)Evaluation experts proposed:“ In the title, is ‘soybean crop’ more appropriate than ‘soybean’?”

A:Thank you for pointing this out,we have changed “soybean”to “soybean crop”.

Title:Estimation of chlorophyll content in soybean crop at different growth stages based on optimal spectral index

(2)Evaluation experts proposed:“In lines 89-91, ‘i.e., the four-node stage (V4), complete flowering stage (R2), full pod stage (R4), and drum-grain stage (R6) was taken as the research object.’Why take these four growth stages as the research object? Are chlorophyll content estimates for other growth stages of soybean not meaningful?”

A:Thank you for pointing this out,these four growth stages almost cover most of the growth process of soybean, and the chlorophyll content of leaves is higher. The rest of the period is seed germination and seedling stage and maturity stage, less leaves or higher degree of aging, less chlorophyll content, little effect on the results and little significance.

(3)Evaluation experts proposed:“ In lines 113-114 and other places, please note the space between the number and unit.”

A:Thank you for pointing this out,we have checked and corrected the spaces between numbers and units.

Line115-116:Four nitrogen application levels were set in this experiment: N0: 0 kg/hm2, N1: 60kg/hm2, N2: 120 kg/hm2, N3: 180 kg/hm2;

(4)Evaluation experts proposed:“ In lines 128-129, I know that SPAD values correlate highly with crop chlorophyll content. However, the SPAD value cannot completely equal to the chlorophyll content. Why not use chemical methods (such as extraction) to determine directly the chlorophyll content of soybean leaves?”

A:Thank you for pointing this out,using chemical methods ( such as extraction ) to directly determine the chlorophyll content of soybean leaves, this method has low requirements for equipment, but the steps are relatively cumbersome, time-consuming and laborious, and the stability of the determination results is not high. The spad-502 chlorophyll content analyzer is used to determine chlorophyll. It is easy to operate and can achieve non-destructive testing. It can track, analyze and study the changes of chlorophyll content in the same leaves. In this study, remote sensing technology is used to estimate chlorophyll content. It is easier to implement and the results are more stable.

(5)Evaluation experts proposed: “In lines 147-149, ‘A total of 63 sets of data were collected for each growth stage, with a total of 252 sets of data collected during the experiment…’ How to get the 63 sets of data for each growth stage? Please explain it in detail.”

A:Thank you for pointing this out,The detailed steps have been supplemented in the paper, see the annotation of the data collection section.

Line141-159: During the test period, spectral data was collected at different soybean growth stages. The test period was sunny and the light was consistent. The spectral reflectance of the soybean canopy was measured by ASD Field-Spec 3 Analytical Spectral Devices, Inc., St, Boulder, CO, USA. The wavelength range of the instrument was 350 -1830 nm. The spectral resolution of 350 - 1000 nm was 3 nm, and the sampling interval was 1.4 nm. The 1000 -1830 nm resolution was 10 nm with a sampling interval of 2 nm, and the instrument automatically interpolates the sampling data to a 1 nm interval output, fiber length of 1.5 m, and field angle of 25 °[15]. Before the hyperspectral data acquisition, the spectrometer was preheated and optimized, and the reference plate test and comparison were completed within 1min.After the hyperspectral reflectance data acquisition of the previous test area, the reference plate correction was performed before the hyperspectral reflectance data acquisition of the latter test area. The crop canopy with balanced growth was selected in each experimental plot, and the experimenter held a spectral sensor probe.The optical fiber probe was placed vertically downward, about 1 m from the top of the canopy. In each plot, three quadrats representing the growth of the crop were selected for measurement. Each quadrat collected ten spectral curves each time and the average value was used as the spectral reflectance of the quadrat.

(6) Evaluation experts proposed: “In lines 154-156, ‘… Savitzky-Golay convolution smoothing was used to preprocess the spectral data.’Can S-G smoothing alone complete so many preprocessing functions, including background noise, baseline drift, and stray light?Please consider other preprocessing methods, such as SNV.”

A:Thanks to the valuable opinions and methods of the expert,The Savitzky-Golay convolution smoothing method can reduce the background noise. The expression in this paper is not accurate. I have corrected it in the corresponding part. The subsequent research will consider the application of SNV method.

Line164-166: To reduce (eliminate) the influence of background noise on the spectral reflectance curve, Savitzky-Golay convolution smoothing was used to preprocess the spectral data.

(7) Evaluation experts proposed: “In lines 178-180. ‘The larger the R2, the higher the prediction accuracy of the model. The smaller RMSE and MRE indicate that the model's performance is more stable, and the prediction results are more concentrated.’I do not think your description is correct.The large R2 value indicates the more stable performance of the trained model, and the prediction results are more concentrated.The smaller RMSE and MRE indicate the higher prediction accuracy of the model.”

A:Thank you for pointing this out,after consulting the literature, we think “The large R2 value indicates the more stable performance of the trained model, and the prediction results are more concentrated.The smaller RMSE and MRE indicate the higher prediction accuracy of the model.”we have corrected it in the corresponding part.

Line225-227: The larger the R2, the more stable performance of the trained model, and the prediction results are more concentrated. The smaller RMSE and MRE indicate the higher prediction accuracy of the model.

(8)Evaluation experts proposed:“ In Figure 2, there is a problem with the structure of the architecture diagram. “chlorophyll content at different growth stages” should not be behind the “Optimal spectral index”, and should be in parallel with ‘Spectral analysis’.”

A:Thank you for pointing this out,we have corrected the architecture diagram.

(9)Evaluation experts proposed:“In lines 197-207, Table 2 and the corresponding description content should be placed in Section 2 Materials and methods.”

A:Thank you for pointing this out,we have placed Table 2 and the corresponding description content in Section 2 Materials and methods.

(10)Evaluation experts proposed:“In line 206, ‘Figures 1 and 2’ are not correct here”

A:Thank you for pointing this out,we have changed it to“Figures 3、4 and 5”.

Line247: As shown in Figures 4、5 and 6, from blue to red, the correlation between the spectral indices and soybean SPAD at each growth stage changes from a high negative correlation to a high positive correlation

(11)Evaluation experts proposed:“ In line 268, ‘… both of which had an excellent linear fitting effect.’

Do SVM and BPNN methods have excellent linear fitting capabilities? Isn't it the nonlinear fitting ability?”

A:Thank you for pointing this out, we have corrected for nonlinear fitting ability.

Line247: The R2 of the modeling set and validation set of the chlorophyll content estimation model, built by the SVM and BPNN methods, exceeded 0.7, both of which had an excellent nonlinear fitting ability.

(12) Evaluation experts proposed:“In Figure 6, please note the number of decimal places.”

A:Thank you for pointing this out,we have checked the number of decimal places and corrected.

(13)Evaluation experts proposed:“ In the Results and Discussion section, the paper emphasizes that the first-order differential spectra contain more useful information, and the best spectral index is selected from the first-order differential spectra.

However, this paper does not perform a comparison between the original spectrum and the first-order differential spectrum.”

A:Thank you for pointing this out, in many previous studies, compared with the original spectrum, the first-order differential treatment is better and contains more useful information. Therefore, this paper directly selects the optimal spectral index in the first-order differential spectrum.

Tang Z J.; Guo J J.; Xiang Y Z.; Lu X H.; Wang Q.; Wang H D.; Cheng M H.; Wang H.; Wang X.; An J Q.; Abdelghany A.; Li Z J.; Zhang F C. Estimation of Leaf Area Index and Above-Ground Biomass of Winter Wheat Based on Optimal Spectral Index[J]. Agronomy-Basel. 2022, 12(7), 1729.Doi: 10.3390/agronomy12071729

Reviewer 3 Report

The article is very promising and presents relevant results, however I believe that several corrections are necessary throughout the text so that the article is suitable for publication. I made my considerations in the attached file. Good luck to the authors

Author Response

Responses to Reviewer #3 (Manuscript ID: Agronomy- 2199247)

Thank you for addressing all my comments.

Thank you for your careful review and positive comments. We have now incorporated the reviewer’s comments and suggestions in preparation of the revised manuscript. I have corrected the grammatical errors and chart formats. The all modified part is marked in red in the manuscript.

(1)“That doesn't correspond to the truth... traditional methods are destructible, but are still the most reliable for parameter quantification. If this were really true, what would be your reference data for the models you proposed?”

A:Thank you for pointing this out,The traditional measurement method is destructive sampling, and the result is unstable. The data in this paper are measured by SPAD-502 instrument.

(2)“this to me seems confusing... do you use spectral information to model vegetation indices??? or do you use spectral information to calculate vegetation indices and model vegetation parameters???”

A:Thank you for pointing this out,We used spectral information to calculate and model the vegetation index.

(3)“beware of words significantly and highly. Analyzing the metrics obtained in the models constructed by the cited authors, these words do not apply.”

A:Thank you for pointing this out,We have made changes.

(4)“What is the main purpose of your work? that was not clear. Here's a hypothesis and the objects of study... but where is the objective?”

A:Thank you for pointing this out. The effects of different growth stages and machine learning methods on the accuracy of the soybean chlorophyll content prediction model were discussed, which could provide a theoretical basis for monitoring the growth and health of soybean crops at different growth stages.

(5)“Wouldn't that be N0?”

A:Thank you for pointing this out. Misrepresentation, modified.

Line122: N0 nitrogen treatment (CK) was the control. Each treatment was randomly arranged and repeated twice.

(6)“please include a small chlorophyll measurement scheme in the collected plant.”

A:Thank you for pointing this out. The measurement method is :Ten soybean plants were randomly selected from each plot at different stages, and the SPAD values of the top 1, top 2, top 3, and top 4 leaves were measured from top to bottom along the main stem. At the same time, for the leaves at the four above leaf positions, from the base of the leaves, according to their length, every 1/3 is divided into intervals which are defined as the base (B), middle (C), and top (R). The SPAD values were measured and the average values in the same experimental plots were calculated.

(7)“how big is this quadrat? the measurements within that quadrat were in different locations?”

A:Thank you for pointing this out. The sample is very small at a certain position of the blade, and the measured values in the sample are basically in the same position.(Line154-159)

(8)“In figure 2,this diagram is not informative and becomes unnecessary.”

A:Thank you for pointing this out. We think this figure is the overall structure of the study, and made the corresponding correction.

(9)For the proposed grammatical errors and chart format problems,We have corrected the grammatical errors and chart formats in this manuscript.

(10)“this belongs to the material and methods and this text is confusing. From what's written it looks like you adjusted the spectrum by first-order treatment, calculated the index, and then correlated with chlorophyll. However, from the above in figures 3, 4 and 5 you made a correlation after the first order treatment, to choose the best spectral band. You need to correct this text.”

A:Thank you for pointing this out.We have corrected the marked words and charts in the manuscript.(Line168-178)

(11)“these matrices are drawn wrong... by your text it is clear that you did the spectral first order treatment, calculated the indices and then correlated with chlorophyll, but here it seems that the correlation was made considering only the spectral information.”

A:Thank you for pointing this out.These matrices are my first-order spectral processing, calculated the index and then related to chlorophyll. The spectral information in the matrix is measured at the leaf level and contains information related to chlorophyll.

(12)“unnecessary, repetitive material and methods”

A:Thank you for pointing this out.For the repetitive and unnecessary parts mentioned in the text, we also revise them one by one.

(13)“Analyze your data... this is not correct. The lowest observed value of RMSE is from stages V4 and R2.”

A:Thank you for pointing this out.The corresponding parts have been changed.(Line303)

(14)“for the same R4 stage of growth?”

A:Thank you for pointing this out. It is the same R4 stage of growth.(Line305-307)

(15)“the text got confused. Present only the validation values of the model to avoid confusion.”

A:Thank you for pointing this out.The text and format of table 4 have been adjusted to an appropriate level.

(16)“choose how you want to present your results, in table 4 or figure 6. It's very repetitive.”

A:Thank you for pointing this out.Table 4 is the specific results of the model, and Figure 7 (Modified number)is the visual map of the data set, which can better present the model results.

(17)“you missed showing the results of the model before the first-order differential correction.again, you didn't show your models without the transformation... it is not clear that the transformation of 1st order was responsible for the best accuracy of its model.”

A:Thank you for pointing this out. In many previous studies, compared with the original spectrum, the first-order differential treatment is better and contains more useful information. Therefore, this paper directly selects the optimal spectral index in the first-order differential spectrum.

Tang Z J.; Guo J J.; Xiang Y Z.; Lu X H.; Wang Q.; Wang H D.; Cheng M H.; Wang H.; Wang X.; An J Q.; Abdelghany A.; Li Z J.; Zhang F C. Estimation of Leaf Area Index and Above-Ground Biomass of Winter Wheat Based on Optimal Spectral Index[J]. Agronomy-Basel. 2022, 12(7), 1729.Doi: 10.3390/agronomy12071729

Reviewer 4 Report

Research on the assessment of chlorophyll in soybeans in different stages of growth using the spectral index is an extremely interesting scientific research that has the potential for further development and wider application.

However, the equations must be checked because equation no. 1 is not correct according to what is currently written - the numerator and the denominator are exactly the same?!

Table 3 and 4. isn't the first column growth phase, not duration?

Table 1: for the observed indices (first column) it is necessary to indicate the units of measurement, and if there are none, this should also be indicated - the same is true for the amount of chlorophyll (first row of the table)

Sincerely,

Author Response

Responses to Reviewer #4 (Manuscript ID: Agronomy- 2199247)

Thank you for addressing all my comments.

Response: Thank you for your careful review and positive comments. We have now incorporated the reviewer’s comments and suggestions in preparation of the revised manuscript. The modified part is marked in red in the manuscript.

(1)Evaluation experts proposed:“However, the equations must be checked because equation no. 1 is not correct according to what is currently written - the numerator and the denominator are exactly the same?!”

A:Thank you for pointing this out,we ' ve checked these equations and now they are all correct.(Line229)

(2)Evaluation experts proposed:“Table 3 and 4. isn't the first column growth phase, not duration?”

A:Thank you for pointing this out, in order to maintain consistency, it is changed to the growth stages.

(3)Evaluation experts proposed:“Table 1: for the observed indices (first column) it is necessary to indicate the units of measurement, and if there are none, this should also be indicated - the same is true for the amount of chlorophyll (first row of the table)”

A:Thank you for pointing this out, for the observed indices (first column) and the amount of chlorophyll (first row of the table),they are all without units, I have marked them all.

Reviewer 5 Report

Three machine learning based approaches, SVM, RF, and BPNN are selected as a prediction model. Estimation of chlorophyll content in soybean is presented with machine learning based approaches and the idea is clear. However, there exits issue to be addressed.

1. Please explain in detail about the limitations of machine learning based approaches used in the study for the justification of superiority.

2. In introduction, It will be better to introduce other novel machine learning and deep learning based approaches that may be superior than classical decision tree based approaches such as RF, and backpropagation neural networks. For instance,

i. Shah, C.; Du, Q.; Xu, Y. Enhanced TabNet: Attentive Interpretable Tabular Learning for Hyperspectral Image Classification. Remote Sens. 202214, 716. https://doi.org/10.3390/rs14030716 

ii. Zheng, H.; Wu, Y. A XGBoost Model with Weather Similarity Analysis and Feature Engineering for Short-Term Wind Power Forecasting. Appl. Sci. 2019, 9, 3019. https://doi.org/10.3390/app9153019

3. It will be better to include some visualization maps for the dataset used in the study.

4. In methodology, It will be better to highlight the main differences of the proposed method in comparison to existing methods.

5. For experiments of methodology, please explain in detail about the hyperparameter tuning of the implemented algorithms. It will be better to illustrate in detail about the validation.

6. It will be better to give some analysis about the complexity of models for better comparison of their performance.

7. Some English writing typos should be corrected.

8. In conclusion, it will be better to explain about the future work implementation relating to the presented work.

Author Response

Responses to Reviewer #5 (Manuscript ID: Agronomy- 2199247)

Thank you for addressing all my comments.

Response: Thank you for your careful review and positive comments. We have now incorporated the reviewer’s comments and suggestions in preparation of the revised manuscript. The modified part is marked in red in the manuscript.

(1)Evaluation experts proposed:“ Please explain in detail about the limitations of machine learning based approaches used in the study for the justification of superiority”

A:Thank you for pointing this out, at present, machine learning methods are widely used, with high efficiency of learning and prediction, easy to implement, and nonlinear fitting ability. The method based on machine learning used in the study is to select several classical machine learning methods to predict the physiological indexes of crops under the condition of unknown results, and to select the optimal machine learning method. Due to the influence of experimental environment, crop varieties and evaluation indexes, the diversity of physiological indexes and the types of machine learning cannot quickly find the optimal combination.

(2)Evaluation experts proposed:“In introduction, It will be better to introduce other novel machine learning and deep learning based approaches that may be superior than classical decision tree based approaches such as RF, and backpropagation neural networks. For instance,

  1. Shah, C.; Du, Q.; Xu, Y. Enhanced TabNet: Attentive Interpretable Tabular Learning for Hyperspectral Image Classification. Remote Sens. 2022, 14, 716. https://doi.org/10.3390/rs14030716 
  2. Zheng, H.; Wu, Y. A XGBoost Model with Weather Similarity Analysis and Feature Engineering for Short-Term Wind Power Forecasting. Appl. Sci. 2019, 9, 3019. https://doi.org/10.3390/app9153019”

A:Thank you for pointing this out, in this paper, we have introduced other new machine learning and deep learning based methods and prospects for the future.(Line376-387)

(3)Evaluation experts proposed:“ It will be better to include some visualization maps for the dataset used in the study.”

A:Thank you for pointing this out, the spectral data and modeling data sets used in the article have been presented in the form of visual maps.

(4)Evaluation experts proposed:“In methodology, It will be better to highlight the main differences of the proposed method in comparison to existing methods.”

A:Thank you for pointing this out, the main differences between the methods proposed in this study and the existing methods have been added and supplemented, as noted in the paper.(Line87-103)

(5)Evaluation experts proposed:“For experiments of methodology, please explain in detail about the hyperparameter tuning of the implemented algorithms. ”

A:Thank you for pointing this out, a detailed introduction to all machine learning methods has been added, including the adjustment of various parameters.(Line186-219)

(6)Evaluation experts proposed:“ It will be better to give some analysis about the complexity of models for better comparison of their performance.”

A:Thank you for pointing this out, a detailed introduction to all machine learning methods has been added, including the adjustment of various parameters, which can show the complexity of different models.(Line186-219)

(7)Evaluation experts proposed:“Some English writing typos should be corrected.”

A:Thank you for pointing this out, we have corrected the typos in English writing.

(8)Evaluation experts proposed:“In conclusion, it will be better to explain about the future work implementation relating to the presented work.”

A:Thank you for pointing this out, we have added the submitted work related to future work implementation.(Line388-396)

Round 2

Reviewer 1 Report

Thank you for making these edits; the manuscript has greatly improved. However, it will be further strengthened and made more impactful if the authors elaborate on these points in the manuscript. Please find additional comments below, in bold-face.

(2)Evaluation experts proposed:“The SPAD-502 meter uses optical densities at 650 nm and 940 nm to estimate chlorophyll, so it is interesting indices using those wavelengths did not have higher correlation coefficients.”

A: Thank you for pointing this out. The optimal band used in some indexes in this paper is the band near 650 nm, which has a certain deviation. After the first-order differential transformation, the results also have corresponding changes, so the correlation coefficient is not so high.

Yes, but why might the reflectance at 940 nm, and indices comparing reflectances at these two bands, have low correlations?

(6)Evaluation experts proposed:“ Since SPAD meter chlorophyll measurements were collected at the leaf level, would it not make more sense to collect ASD spectra with a 1-degree attachment rather than the bare fiber optic with a 25-degree FOV?”

A:Thank you for pointing this out, in the growth period of soybean, when the field angle is 25 °, the measurement height of 100 cm can better obtain the hyperspectral reflectance of soybean. In the current experimental environment and measurement height, the 25-degree optical fiber probe can collect the spectral data of crop leaves more easily and comprehensively, which is more suitable for my research content.

Yes, these benefits of using the bare fiber FOV are valid; however, I would still recommend discussing the impacts of using this versus the 1-degree attachment on your results in the discussion section.

(8)Evaluation experts proposed:“For SVM, why was the radial basis function not used, since it can handle non-linearity better?”

A:Thank you for pointing this outSVM uses Gaussian kernel and polynomial kernel as basis kernel function, Gaussian kernel function can map a sample to a higher dimensional space, the kernel function is the most widely used, whether large sample or small sample has better performance ; the polynomial kernel makes the original linear inseparable data linearly separable, which can solve the nonlinear problem, and has good adaptability and application results for the kernel function in this study.

Yes, SVM can use Gaussian and polynomial kernels, but I believe other commonly used kernel functions would lead to better performance. I still recommend re-running the analyses with other kernels and seeing whether accuracies improve.

(11)Evaluation experts proposed:“For the spectral indices listed in Table 2, please elaborate on why the reflectances at 455 nm and 550 nm were used.”

A:Thank you for pointing this outin previous studies, the spectral reflectance at 445 nm and 550 nm wavelength positions was selected for the calculation of spectral index, and good results were obtained. Therefore, the spectral reflectance at 445 nm and 550 nm wavelength positions was also selected in this study.

Román J R, Rodríguez-Caballero E, Rodríguez-Lozano B,et al. Remote Sensing, 2019,11(11):1.

Sun H, Feng M C, Xiao L J, et al.PLOS ONE,2019,14(6):e0216890.

Lu J Z, Ehsani R, Shi Y Y.Scientific Reports,2018,8:2793.

Liu Shuang, Yu Haiye, Zhang Junhe, Zhou Haigen, Kong Lijuan, Zhang Lei, Dang Jingmin, Sui Yuanyuan. Research on inversion model of soybean leaf chlorophyll content based on optimal spectral index [ J ].Spectroscopy and spectral analysis, 2021,41 ( 06 ) : 1912-1919.

Yes, please add this information and citations to the text.

(12)Evaluation experts proposed:“Also in Table 2, there are so many chlorophyll-specific indices (Gitelson Chl index, Modified Chlorophyll Absorption in Reflectance Index, Triangular Greenness Index, Normalized Pigment Chlorophyll Index, etc). Why were these not included?”

A:Thank you for pointing this outRatio Index ( RI ), Difference Index ( DI ), Modified Simple Ratio ( mSR ), Modified Normalized Difference Index ( mNDI ), Triangular Vegetation Index ( TVI ), Soil-Adjusted Vegetation Index ( SAVI ) and Normalized Difference Vegetation Index ( NDVI ) are seven typical spectral indices related to chlorophyll content. In addition, it has been widely used in previous studies, and the effect is better, so this study also uses these indexes, and will consider the substitution of other indexes in future studies.

Yes, please add references to these previous studies. Also, please add descriptions of the other indices that could be used in the discussion section.

(13)Evaluation experts proposed:“It is interesting that DI outperformed NDVI since the latter is so often used to describe “greenness.” Please elaborate on why this might have happened.”

A:Thank you for pointing this outAlthough NDVI is often used to describe“greenness.”, DI may be more correlated with chlorophyll content at some stages due to factors such as the test environment and crop species.

Yes, please add this explanation to the manuscript.

(17)Evaluation experts proposed:“Since correlations were strongest in R4, is this also the best stage for farmers to estimate soybean yield?”

A:Thank you for pointing this outin the R4 growth stage, the correlation between spectral index and chlorophyll content was the highest, and the chlorophyll content was closely related to the yield, showing a positive correlation. This study can provide a theoretical basis for farmers to predict yield in crop production, that is, R4 period can be the best period for farmers to predict yield.

Yes, please add this and other implications and applications of these findings in the discussion section, as this will strengthen the manuscript.

(21)Evaluation experts proposed:“Also in Figure 6, we see observed chlorophyll values are often higher than estimated values. This might be because of the differences in field of view of the SPAD meter (leaf level) and the ASD (plant level).”

A:Thank you for pointing this outIn Fig.6, the observed chlorophyll values are often higher than the estimated values. The influence of multiple factors in the experiment is inevitably affected by errors. The deviation of instrument and measurement angle may affect the results. In addition, as experts said, the different field of vision of SPAD measuring instrument ( leaf level ) and ASD ( plant level ) will also lead to differences between the observed value and the estimated value.

Yes, please add these details to the discussion.

(22)Evaluation experts proposed:“Please describe the canopy closure and/or senescent levels in each of the growth stages. How much influence might there be from the soil background and/or dead vegetation?”

A:Thank you for pointing this outin the four-node stage(V4), Soybean is in the earlier growing stage, the vegetation area is moderate, and a small amount of soil is exposed. In complete flowering stage (R2), full pod stage (R4) and drum-grain stage (R6),The vegetation is dense and almost completely covers the ground.The degree of aging gradually deepened.The soil background was eliminated in the spectral treatment, which had little effect on the results. In addition, due to better field management, the crop grew well, no pests and diseases, almost no dead vegetation, so it did not affect the experimental results.

Yes, please elaborate on this in the manuscript as it will help readers interpret your results. How was the soil background removed? The SG smoothing would not remove the soil signature. Also, was there crop residue on the ground that might influence the spectra? Is it correct that plants in the R6 stage have not begun senescing? These details help readers assess whether the results make sense.

(23)Evaluation experts proposed:“Usually, information on crop vigor is obtained by comparing reflectances at the red and near-infrared bands, rather than only the red edge. Please elaborate on why the red edge bands were most important in this study.”

A:Thank you for pointing this outthe strong absorption of chlorophyll in the red band and the strong reflection of the near-infrared band inside the leaves make the red-edge band the fastest growth of green plant reflectance and the most important indicator band that can reflect the physiological characteristics of plant growth. More than 80 % of the physical and chemical parameters of plants can be mapped from the spectral information contained in the red-edge band. Therefore, the red edge band is the most important in this study.

Yes, please add this to the manuscript.

Also, please double-check spacing, spelling, and grammar in the added text (e.g., nrural).

In Table 1, please change “none units” to “Unitless” or “No units.” Also, why are V4 and R2 bold-faced but not R4 and R6?

Author Response

Responses to Reviewer #1 (Manuscript ID: Agronomy- 2199247)

Thank you for addressing all my comments.

Thank you for your careful review and positive comments. We have now incorporated the reviewer’s comments and suggestions in preparation of the revised manuscript. The modified part is marked in red in the manuscript.

(1)Evaluation experts proposed:“ Yes, but why might the reflectance at 940 nm, and indices comparing reflectances at these two bands, have low correlations?”

A : Thank you for pointing out that the 940 nm spectral band did not reach the optimal band due to the influence of water and nitrogen stress and experimental environment due to the different water and nitrogen treatment in the experiment, and its correlation was not as high as the current selected band.

(2)Evaluation experts proposed:“Yes, these benefits of using the bare fiber FOV are valid; however, I would still recommend discussing the impacts of using this versus the 1-degree attachment on your results in the discussion section.”

A: Thank you for pointing this out. In the soybean growth period, when the field angle is 25 °, the measurement height of 100 cm can better obtain the hyperspectral reflectance of soybean. In the current experimental environment and measurement height, the 25 degree optical fiber probe can collect the spectral data of crop leaves more conveniently and comprehensively, which is more suitable for my research content, and most of the previous studies used the 25 degree optical fiber probe to measure the spectral data. In the experiment, the data was collected by the 1 degree optical fiber probe, and the results were relatively good, but there was no literature research with the 1 degree optical fiber probe, so it was not used. In future research, we will consider using a 1 degree optical fiber probe measurement and explore its results and effects.

(3)Evaluation experts proposed:“Yes, SVM can use Gaussian and polynomial kernels, but I believe other commonly used kernel functions would lead to better performance. I still recommend re-running the analyses with other kernels and seeing whether accuracies improve.”

A:Thank you for pointing this out, We have used the SVM radial basis kernel function to process the current data ( for example, in the R4 period, the R2 of the validation set is 0.886, RMSE = 2.394, MRE = 4.219 ), and found that the fitting degree and accuracy have been improved to varying degrees. However, the kernel function currently used is more efficient and easy to use for us. In future research, we will consider the new kernel function to optimize the model and strive to get better results.

(4)Evaluation experts proposed:“ Yes, please add this information and citations to the text.”

A:Thank you for pointing this out, we added this information and citations in the text.

(5)Evaluation experts proposed:“Yes, please add references to these previous studies. Also, please add descriptions of the other indices that could be used in the discussion section.”

A:Thank you for pointing this out, we have referred to previous studies and added an explanation of the index in the corresponding section.

(6)Evaluation experts proposed:“Yes, please add this explanation to the manuscript.”

A:Thank you for pointing this out, we added this explanation in the text.

(7)Evaluation experts proposed:“ Yes, please add this and other implications and applications of these findings in the discussion section, as this will strengthen the manuscript.”

A:Thank you for pointing this out,we add this and other implications and applications of this findings in the discussion section.

(8)Evaluation experts proposed:“Yes, please add these details to the discussion.”

A:Thank you for pointing this out,we added these details to the discussion.

(9)Evaluation experts proposed:“Yes, please elaborate on this in the manuscript as it will help readers interpret your results. How was the soil background removed? The SG smoothing would not remove the soil signature. Also, was there crop residue on the ground that might influence the spectra? Is it correct that plants in the R6 stage have not begun senescing? These details help readers assess whether the results make sense.”

A:Thank you for pointing this out,At the beginning, the binary Ostu algorithm and Canny edge detection algorithm are used to process the thermal infrared image in advance to eliminate the soil background. Due to previous negligence, not described in the text, has been added. In addition, due to better field management, there will be no crop residues on the ground that may affect the spectrum. In the R6 stage, the crop begins to age, but the field coverage is still good, no pests and diseases, almost no dead vegetation, so it is not affected.

(10)Evaluation experts proposed:“ Yes, please add this to the manuscript.”

A:Thank you for pointing this out,we added this in the manuscript.

(11)Evaluation experts proposed:“Also, please double-check spacing, spelling, and grammar in the added text (e.g., nrural). In Table 1, please change “none units” to “Unitless” or “No units.” Also, why are V4 and R2 bold-faced but not R4 and R6?”

A:Thank you for pointing this out. We have carefully checked spacing, spelling and grammar in the added text. In Table 1, we have changed ' none unit ' to ' Unitless '. At the same time, we have thickened both R4 and R6.

Reviewer 2 Report

Dear author

I think that some questions can be further solved.

1. In line 124, this paper indicates 33 plots.

Lines 155-159 indicate that: "In each plot, three quadrats representing the growth of the crop were selected for measurement. Each quadrat collected ten spectral curves each time and the average value was used as the spectral reflectance of the quadrat. A total of 63 sets of data were collected for each growth stage, with a total of 252 sets of data collected during the experiment (Table 1)."

However, I am still confused about the 63 samples for each growth stage.

2. Lines 225-227: The larger the R2, the more stable performance of the trained model, and the prediction results are more concentrated. The smaller RMSE and MRE indicate the higher prediction accuracy of the model.

Herein, it is recommended to cite a paper. For example, https://doi.org/10.3390/rs12172826

3. In Figure 7, there are too many decimal places. Three decimal places are recommended.

4. Authors have replied that "These four growth stages almost cover most of the growth process of soybean, and the chlorophyll content of leaves is higher. The rest of the period is seed germination and seedling stage and maturity stage, with fewer leaves or a higher degree of aging, less chlorophyll content, little effect on the results, and little significance."

I suggest that the above content is added to the second paragraph of the Introduction to explain why the four-node stage (V4), complete flowering stage (R2), full pod stage (R4), and drum-grain stage (R6) were taken as the research object.

Author Response

Responses to Reviewer #2 (Manuscript ID: Agronomy- 2199247)

Thank you for addressing all my comments.

Thank you for your careful review and positive comments. We have now incorporated the reviewer’s comments and suggestions in preparation of the revised manuscript. The modified part is marked in red in the manuscript.

(1)Evaluation experts proposed:“ However, I am still confused about the 63 samples for each growth stage.”

A:Thank you for pointing this out,There are 33 plots in the experimental area. Spectral data were collected for each plot in each growth period, and a total of 66 sets of data were collected for two consecutive years. Due to the irresistible factors such as vegetation death, outliers need to be eliminated. However, due to the modeling and prediction, the ratio of training set to validation set is 2 : 1, so in order to facilitate modeling, three related outliers need to be eliminated, and finally there are 63 sets of data in each growth period.

(2)Evaluation experts proposed:“Lines 225-227: The larger the R2, the more stable performance of the trained model, and the prediction results are more concentrated. The smaller RMSE and MRE indicate the higher prediction accuracy of the model. Herein, it is recommended to cite a paper. For example, https://doi.org/10.3390/rs12172826”

A:Thank you for pointing this out. Here we have added cited papers.

(3)Evaluation experts proposed:“ In Figure 7, there are too many decimal places. Three decimal places are recommended.”

A:Thank you for pointing this out,we have modified Figure 7 and left the result to 3 decimal places.

(4)Evaluation experts proposed:“I suggest that the above content is added to the second paragraph of the Introduction to explain why the four-node stage (V4), complete flowering stage (R2), full pod stage (R4), and drum-grain stage (R6) were taken as the research object.”

A:Thank you for pointing this out,we have added the above to the Introduction.

Reviewer 3 Report

The text has been significantly improved and I wish you good luck in publishing.

Author Response

Responses to Reviewer #3 (Manuscript ID: Agronomy- 2199247)

Thank you for addressing all my comments and blessing.

Thank you for your careful review and positive comments. We have now incorporated the reviewer’s comments and suggestions in preparation of the revised manuscript.

Reviewer 5 Report

The authors have addressed all of my concerns. It looks ready for publication.

Author Response

Responses to Reviewer #5 (Manuscript ID: Agronomy- 2199247)

Thank you for addressing all my comments and blessing.

Thank you for your careful review and positive comments. We have now incorporated the reviewer’s comments and suggestions in preparation of the revised manuscript.